# Cell-type specific innervation of cortical pyramidal cells at their apical dendrites

**Ali Karimi[†], Jan Odenthal[†], Florian Drawitsch, Kevin M Boergens, Moritz Helmstaedter\***

Department of Connectomics, Max Planck Institute for Brain Research, Frankfurt, Germany

**Abstract** We investigated the synaptic innervation of apical dendrites of cortical pyramidal cells in a region between layers (L) 1 and 2 using 3-D electron microscopy applied to four cortical regions in mouse. We found the relative inhibitory input at the apical dendrite's main bifurcation to be more than 2-fold larger for L2 than L3 and L5 thick-tufted pyramidal cells. Towards the distal tuft dendrites in upper L1, the relative inhibitory input was at least about 2-fold larger for L5 pyramidal cells than for all others. Only L3 pyramidal cells showed homogeneous inhibitory input fraction. The inhibitory-to-excitatory synaptic ratio is thus specific for the types of pyramidal cells. Inhibitory axons preferentially innervated either L2 or L3/5 apical dendrites, but not both. These findings describe connectomic principles for the control of pyramidal cells at their apical dendrites and support differential computational properties of L2, L3 and subtypes of L5 pyramidal cells in cortex.

**\*For correspondence:**
mh@brain.mpg.de

[†]These authors contributed equally to this work

## Introduction

The apical dendrites of pyramidal neurons, the most abundant cell type in the mammalian cerebral cortex (*Cajal, 1899*; *DeFelipe and Fariñas, 1992*), have been a key focus of anatomical (*Cajal, 1899*), electrophysiological (*Stuart and Sakmann, 1994*; *Larkum et al., 1999*) and in-vivo functional studies (*Svoboda et al., 1997*; *Letzkus et al., 2011*; *Takahashi et al., 2016*). Of particular interest has been the coupling of long-range top-down inputs converging at the apical tufts in layer 1 with synaptic inputs on the basal dendrites in large layer 5 pyramidal cells (*Larkum et al., 1999*; *Larkum, 2013*). Since almost all pyramidal cells from cortical layers 2, 3 and 5 (and some from layer 4) form their apical dendrite's main bifurcation in this region, the layer 1/2 border in the upper cortex allows for synaptic convergence onto 80–90% of all cortical pyramidal cells (*Larkman and Mason, 1990*; *Ito et al., 1998*). Furthermore, the apical dendrites' main bifurcation has been implied as a site of origin and control of regenerative electrical events in the distal dendrites (*Helmchen et al., 1999*; *Larkum and Zhu, 2002*; *Larkum et al., 2004*). Understanding the innervation profile of synaptic input onto apical dendrites in this peculiar part of the mammalian cerebral cortex was the goal of this study.

## Results

We acquired four 3D-EM datasets from mouse somatosensory (S1), secondary visual (V2), posterior parietal (PPC) and anterior cingulate cortex (ACC) sized between $72 \times 93 \times 141\ \mu m^3$ and $66 \times 89 \times 202\ \mu m^3$ (*Figure 1a*, *Figure 1—figure supplement 1*, Table 1 in *Supplementary file 1*) at a voxel size of $11.24–12 \times 11.24–12 \times 28–30\ nm^3$ using serial block-face electron microscopy (SBEM [*Denk and Horstmann, 2004*]). The image volumes were located at the border of layers 1 and 2, the site of the main bifurcation of apical dendrites from almost all pyramidal neurons of layers 2 to 5 (*Figure 1a,b*). To determine the layer origin of each apical dendrite (AD), we first identified those ADs with a soma in the image volume as L2 ADs, and the remaining ADs as 'deeper layer' (DL ADs),

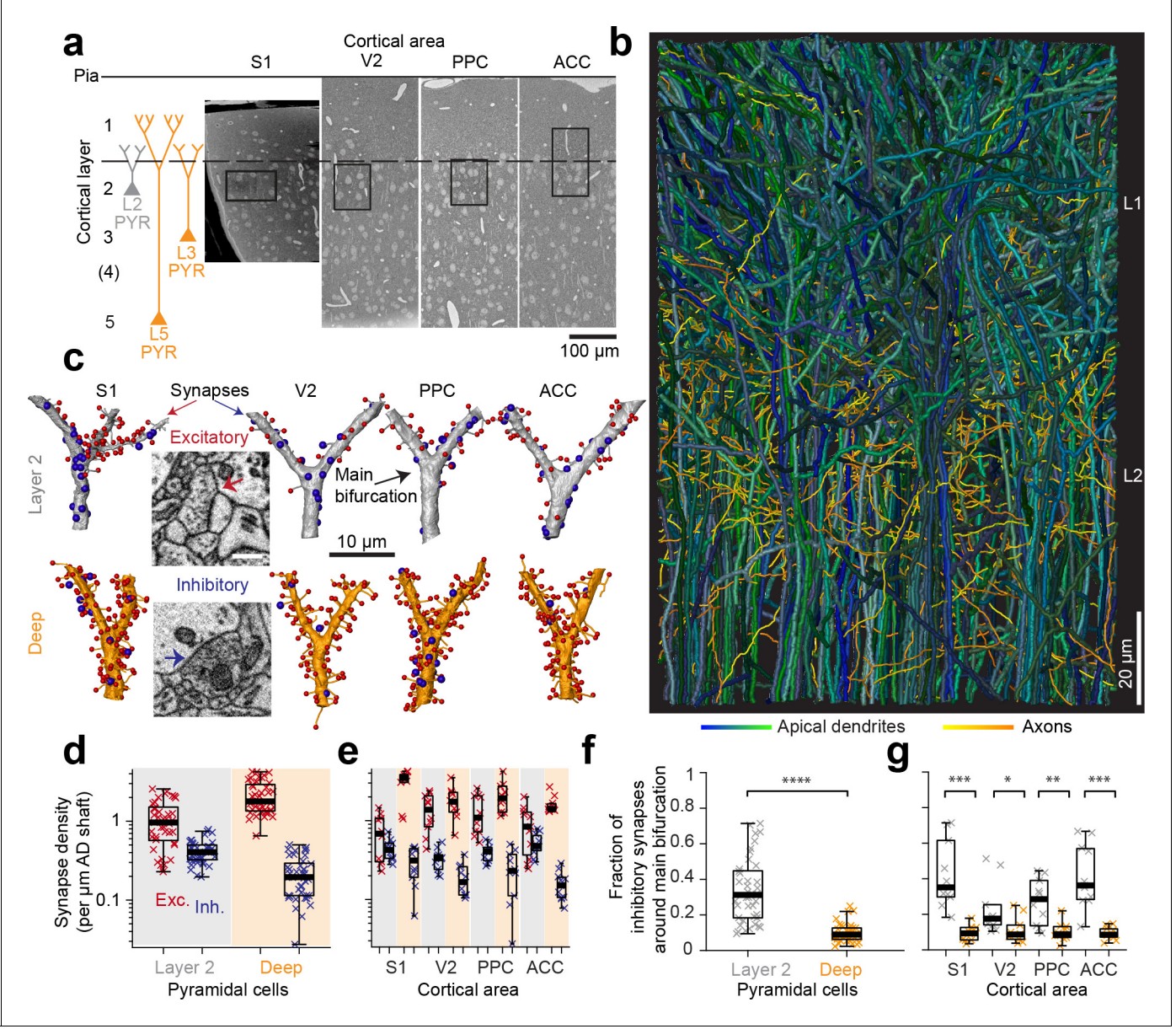

**Figure 1.** Complete synaptic input mapping of pyramidal cell apical dendrites (ADs) around their main bifurcation. (**a**) Overview EM images indicating the location of 3D EM datasets in primary somatosensory (S1), secondary visual (V2), posterior parietal (PPC) and anterior cingulate (ACC) cortices relative to pia surface (solid line) and layer 1 / 2 border (dashed line). Schematic location of layer 2 (gray) and deep layer (orange) pyramidal neurons. (**b**) Reconstruction of all ADs contained in the ACC dataset (blue-green, n = 61 layer 2 and n = 152 deep layer ADs, respectively) and a subset of axons innervating them (n = 62, yellow-orange). Note that 80–90% of all pyramidal cells in cortex extend their apical dendrites into the L1/2 border region, allowing for massive synaptic convergence (https://wklink.org/8300). (**c**) Complete synaptic input maps of apical dendrite main bifurcations for deep layer (orange, S1: https://wklink.org/5413, V2: https://wklink.org/6059, PPC: https://wklink.org/9261, ACC: https://wklink.org/8408) and layer 2 (gray, S1: https://wklink.org/4049, V2: https://wklink.org/8965, PPC: https://wklink.org/8307, ACC: https://wklink.org/8695) pyramidal cells (example excitatory (spine, red spheres, https://wklink.org/9524) and putative inhibitory (shaft, blue spheres, https://wklink.org/2603) synapses, inset). (**d**) Box plot of putative inhibitory (blue crosses) and excitatory (red crosses) synapse densities per μm of AD shaft path length for L2 (n = 41, left) and DL ADs (n = 41, right) in S1, V2, PPC and ACC datasets. Wilcoxon rank-sum test, $p < 10^{-6}$, $10^{-7}$ for excitatory and putative inhibitory densities, respectively. (**e**) Same as (**d**) reported separately per cortical region (n = 20 for S1, V2 and PPC, n = 22 for ACC). (**f**) Box plot of fraction of putative inhibitory synapses at the main bifurcation of deep (orange) and layer 2 (gray) ADs; individual ADs shown (crosses). (**g**) As (**f**) reported separately per cortical region. Asterisks indicate significance level of the Wilcoxon rank-sum test (*$p < 0.05$, **$p < 0.01$, ***$p < 10^{-3}$, ****$p < 10^{-4}$). Scale bars: 0.5 μm (inset in c).

The online version of this article includes the following source data and figure supplement(s) for figure 1:

**Source data 1.** The density of excitatory and putative inhibitory synapses (*Figure 1d, e*).

**Source data 2.** The fraction of excitatory and putative inhibitory synapses (*Figure 1f, g*).

*Figure 1 continued on next page*

*Figure 1 continued*

**Figure supplement 1.** Location of cortical tissue used for EM volume imaging.

that is L3 and L5 ADs (*Figure 1a*). Synapses onto ADs were identified as targeting the shafts vs. the spines of ADs (for an identification of shaft-preferring axons as putative inhibitory vs. spine-preferring axons as putative excitatory, see below) without consideration of the symmetry of pre- and postsynaptic densities (*Gray, 1959*; *Colonnier, 1968*).

## Putative inhibitory to excitatory synapse ratio

We then identified all the synapses onto 82 ADs within 10 µm around their main bifurcation (*Figure 1c*, n = 6240 synapses total, of these n = 1092 shaft synapses, total dendritic length analyzed: 3.29 mm). The density of shaft synapses on DL ADs was 0.22 ± 0.02 synapses per µm dendritic shaft path length (mean ± SEM, n = 41, *Figure 1d,e*). Together with the substantial synaptic input to the spines of these ADs (*Figure 1d,e*, 2.16 ± 0.16 synapses onto spines per µm dendritic shaft path length), this yielded a fraction of 9.9 ± 5.1% (*Figure 1f,g*, mean ± SD, n = 41) shaft synapses at the main bifurcations of DL ADs in S1, V2, PPC and ACC cortex.

Next, we analyzed the synaptic input to L2 pyramidal cells' ADs (*Figure 1c–g*). Here, unlike for DL ADs, we found substantial synaptic input to the shafts of the AD at the main bifurcation (*Figure 1d, e*, 0.42 ± 0.02 synapses per µm dendritic shaft path length, mean ± SEM, n = 41), with a fraction of 33.6 ± 17.9% (*Figure 1f,g*, mean ± SD, n = 41) of shaft synapses for S1, V2, PPC and ACC. Thus, L2 ADs receive about 3-fold more relative synaptic input at the shafts of their main bifurcation compared to L3 and L5 pyramidal cells (9.9% vs 33.6%, Wilcoxon rank-sum test, p < $10^{-11}$).

## Shaft vs. spine synapses

For this analysis, we identified putative inhibitory versus excitatory synapses by their location on dendritic shafts versus spines (See Materials and methods, [*Braitenberg and Schüz, 1998*; *Kubota et al., 2016*]). To estimate the resulting rate of misidentification (i.e. of synapses onto shafts that were made by excitatory axons and synapses onto spines that were made by inhibitory axons, *Figure 2a–c*), we selected a subset of synapses onto shafts and spines of ADs and reconstructed the presynaptic axons. Then, we identified all other output synapses of these axons and determined the fraction of axonal synapses made onto spines (*Figure 2a,b*, Table 4 in *Supplementary file 1*; only single spine innervations were considered; analysis separately for synapses made within L1 and L2, *Figure 2b*). We found that, in fact, the preference of axons to either innervate shafts or spines of ADs was almost binary (*Figure 2b*): 91 of 92 axons seeded from the shafts of L2 ADs made at least 80% of their other output synapses again onto shafts (and 89 of 92 axons made at least 90% of their other synapses again onto shafts). Similarly, 34 of 35 axons seeded from spines at L2 ADs made at least 80% of their other synapses as primary spine innervations. Numbers for synapses seeded at DL ADs were comparable. This clear dichotomy supported the identification of spine-preferring axons as putative excitatory and shaft-preferring as putative inhibitory (*Figure 2b*, see also *Motta et al., 2019*). With the exception of axons innervating L5 slender tufted pyramidal cells, the resulting prediction accuracy of single synapses to predict the shaft or spine preference of the presynaptic axon was above 75%, with most innervations above 90% (*Figure 2c*). We used the ensuing correction factor for L5st neurons (*Figure 2c*) in the following.

## Synapse size, multiple spine innervations

We next investigated whether the observed difference in putative inhibitory synapse density for L2 versus DL ADs could be counteracted by differences in synaptic size (shown to be correlated to synaptic strength for excitatory synapses, [*Cheetham et al., 2014*; *de Vivo et al., 2017*; *Holler-Rickauer et al., 2019*, *Figure 2d*]). Synapse size onto spines was indistinguishable between L2 and DL ADs (*Figure 2d*), but synapses onto shafts of L2 ADs were slightly larger than those onto shafts of DL ADs (*Figure 2d*). Thus, if synapse size of shaft synapses was an indicator of synaptic strength, this would further enhance, not compensate the observed difference in putative inhibitory synapse density for L2 vs DL ADs.

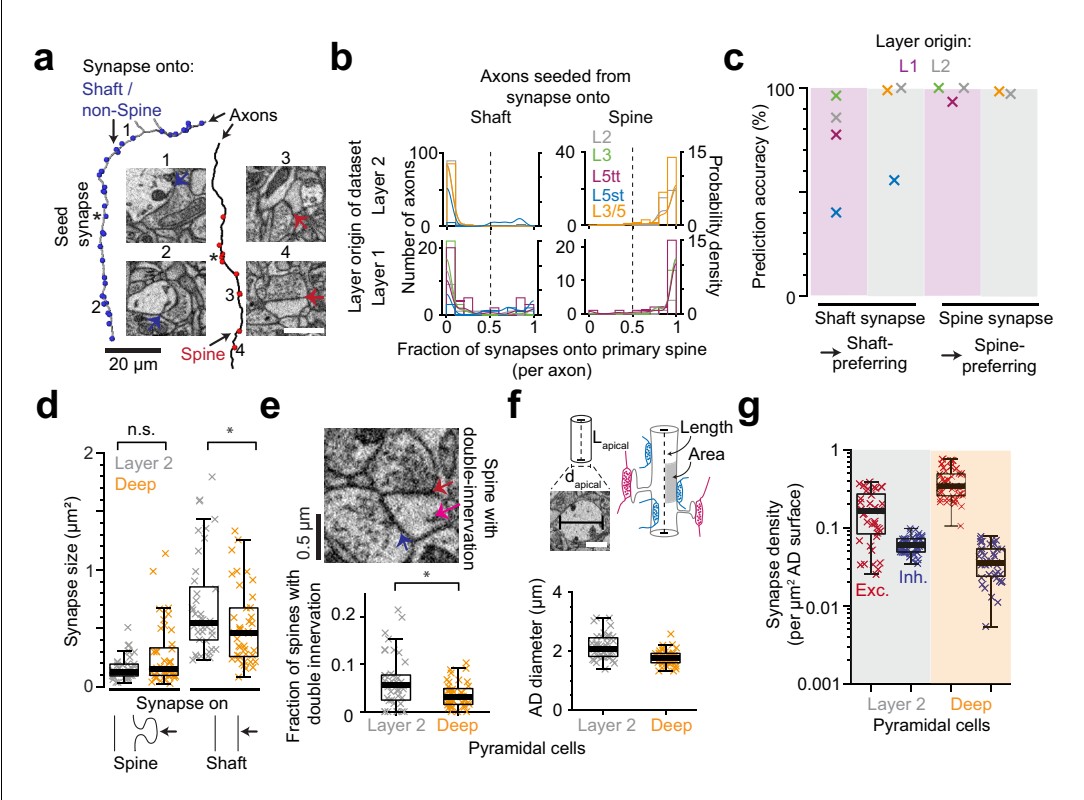

**Figure 2.** Effect of synapse identity, size, dendrite diameter and double-innervated spines on synaptic composition of apical dendrites. (**a**) Reconstruction of two axons seeded from synapses onto ADs (asterisks) and all the output synapses within the volume (red: spine, blue: shaft/non-spine). Note the stark difference in their innervation preference based on their seed location. Example spine (red arrows, 3: https://wklink.org/5224, 4: https://wklink.org/9411) and shaft (blue arrows, 1: https://wklink.org/8807, 2: https://wklink.org/3270) synapse EM micrographs (insets). (**b**) Histogram for the fraction of synapses made onto single-innervated spines. Axons seeded from synapses on deep (orange), L2 (gray), L3 (green), L5tt (magenta) and L5st (blue) ADs. Annotations are from layer 1 (n = 142, LPtA dataset) and layer 2 (n = 288, S1, V2, PPC and ACC datasets); probability density estimations and synapse identity threshold as solid and dashed lines, respectively. Note highly bimodal distribution allowing the clear distinction of axons into spine-preferring (excitatory) and shaft-preferring (putative inhibitory), with exception of axons innervating shafts of L5st ADs (these were corrected in the following, see c). (**c**) Scatter plot for accuracy in predicting presynaptic axon type (shaft- vs. spine-prefering) based on the location of synapse on AD (shaft vs. spine). (**d**) Size of synapses onto deep (n = 41, orange crosses) and layer 2 (n = 41, gray crosses) ADs. (**e**) Fraction of double-innervated spines in the main bifurcation annotations of deep (orange, n = 41) and layer 2 (gray, n = 41) neurons. Spine with putative inhibitory (blue arrow) and excitatory (red arrow) inputs (inset). (**f**) Sketches illustrating the synapse density normalized to surface area and length of AD and their measurement (upper panel). Box plot of the average diameter of layer 2 (gray, n = 41) and deep (orange, n = 41) AD around the main bifurcation in S1, V2, PPC and ACC datasets (lower panel, $p < 10^{-4}$, Wilcoxon rank-sum test). (**g**) Box plot of putative inhibitory (blue crosses) and excitatory (red crosses) synapse densities per $\mu m^2$ of AD surface area for L2 (n = 41, left) and DL ADs (n = 41, right). Wilcoxon rank-sum test, $p < 10^{-6}$ for both densities. Asterisks indicate significance level of the Wilcoxon rank-sum test (not significant (n.s.): $p > 0.05$, *$p < 0.05$). Scale bars: 1 $\mu m$ (inset in a,f).

The online version of this article includes the following source data and figure supplement(s) for figure 2:

**Source data 1.** Dendritic spine vs. shaft innervation preference for AD-targeting axons (***Figure 2b***).

**Source data 2.** Prediction accuracy for dendritic shaft vs. spine innervation preference (***Figure 2c***).

**Source data 3.** Size of synapses onto ADs (***Figure 2d***).

**Source data 4.** Fraction of dendritic spines with double-innervation (***Figure 2e***).

**Source data 5.** Apical dendrite diameter and the synapse density normalized to surface area of ADs (***Figure 2f, g***).

**Figure supplement 1.** Relationship of synapse density to apical dendrite diameter in layers 1 and 2.

A fraction of dendritic spines in cortex have been reported to be innervated by two or more synapses, with the secondary innervation often inhibitory (***Kubota et al., 2007***). A high rate of such multiple spine innervations could thus confound our results (***Figure 1d–g***). We measured the rate of multiple synapses onto spines (***Figure 2e***) and found it to be low (6.4 ± 0.9% vs 3.5 ± 0.4% for L2 vs DL ADs, mean ± SEM, p = 0.014, Wilcoxon rank-sum test). The difference in multiple spine

innervations for L2 vs DL ADs would again further enhance, not reduce, the observed difference in putative inhibitory synapse density for L2 vs DL ADs.

## Synaptic density per dendritic path length vs. surface area

We quantified synapse density per path length of the innervated dendritic shaft of the ADs (*Figure 1*). Differences in dendritic diameter between L2 and DL ADs could yield a differential effect on synaptic density per dendritic surface area for these innervations. It has been argued that areal synapse density is biophysically more relevant (since the input impedance of the dendrite is inversely correlated with its diameter [*Rall and Rinzel, 1973*; *Rinzel and Rall, 1974*; *Bloss et al., 2016*]). We therefore measured the diameter of ADs at their main bifurcation and found L2 ADs to be 21% wider than DL ADs (*Figure 2f*, *Figure 2—figure supplement 1*, diameter of 2.16 ± 0.06 µm vs 1.77 ± 0.04 µm for L2 vs DL ADs, respectively, mean ± SEM, n = 82, Wilcoxon rank-sum test, $p < 10^{-4}$). L2 neurons, even though they had larger diameters, still received 59% higher absolute putative inhibitory synapse density per unit surface area compared to their DL counterparts (*Figures 1d* and *2g*, compare to 89% density difference when normalized to shaft path length). The ratio of putative inhibitory to excitatory synaptic inputs (*Figure 1*) is not affected by these considerations.

## Preference of presynaptic axons for types of apical dendrites

We next wanted to understand whether the presynaptic inhibitory axons that innervated ADs showed any innervation selectivity regarding the layer origin of the targeted pyramidal cells' AD (*Figure 3a–b*). This was of special interest since the main bifurcations of all pyramidal cells reside in the same spatial region at the L1/2 border in cortex (*Figure 3c*, see also *Figure 1b*). We therefore reconstructed presynaptic inhibitory axons that had made at least one synapse onto a pyramidal apical dendrite shaft (*Figure 3a*), identified all other output synapses of these axons, and determined whether these were made again onto the same or different types of pyramidal cell ADs.

About 20% of synapses formed by each axon were again established onto apical dendrites (*Figure 3b,d*, n = 183, 20.3 ± 1.1%, mean ± SEM). AD innervation showed substantial conditional dependence on the type of AD the axon had been seeded from (*Figure 3b*), rejecting a model of indiscriminate inhibitory AD innervation: Axons seeded at DL ADs made 15.4 ± 1.6% of the remaining output synapses onto DL ADs (i.e. 71.2 ± 3% of all synapses onto ADs, mean ± SEM), but only 5.2 ± 0.8% onto L2 ADs (i.e. 28.8% of AD synapses, *Figure 3b*, n = 91, $p < 10^{-8}$, Wilcoxon rank-sum test). Conversely, axons seeded at L2 ADs did not innervate their DL counterparts to a substantial fraction (*Figure 3b*, 4.1 ± 0.6%, n = 92); Rather, they innervated L2 ADs (15.9 ± 1.4%, i.e. 77.4% of their AD synapses). Since the paths of these inhibitory axons along the depth of the cortex were indistinguishable for those axons targeting deep and layer 2 ADs (*Figure 3c*, 18.1 vs. 16.7 mm, n = 183), this connectomic specificity far exceeded average random innervation determined by the trajectory of the presynaptic axons.

## Contribution of multiple innervations to axonal target preference

Next, we investigated to what degree the innervation of individual apical dendrites by multiple synapses from the same inhibitory axon contributed to the establishment of axonal target preference for types of ADs (*Figure 3e*). Axons targeted ADs mostly through monosynaptic innervation regardless of the AD type (*Figure 3f*, 77% of connections made by one synapse; average of 1.32 ± 0.05 vs. 1.29 ± 0.04 synapses per AD target for axons seeded from L2 and DL ADs, respectively, mean ± SEM, Wilcoxon rank-sum test, p = 0.83). Thus, when evaluating the conditional innervation preference of L2 and DL ADs for presynaptic axons while disregarding multiple innervations of the same individual AD by the same presynaptic axon (using a binarized innervation, *Figure 3g*), the conditional axonal target preference was still substantial (*Figure 3g*, compare to *Figure 3b*, 70.8 ± 3.6% vs. 35.3 ± 4.2% individual L2 target fraction for axons seeded from L2 and DL ADs, respectively, mean ± SEM, n = 159, Wilcoxon rank-sum test, $p < 10^{-7}$).

## Comparative analysis of innervation preference between cortex types

Since we had acquired four 3D EM datasets from different parts of the cerebral cortex including primary sensory (S1), secondary sensory (V2), and higher-order (ACC, PPC) areas (*Figure 4a*), we then

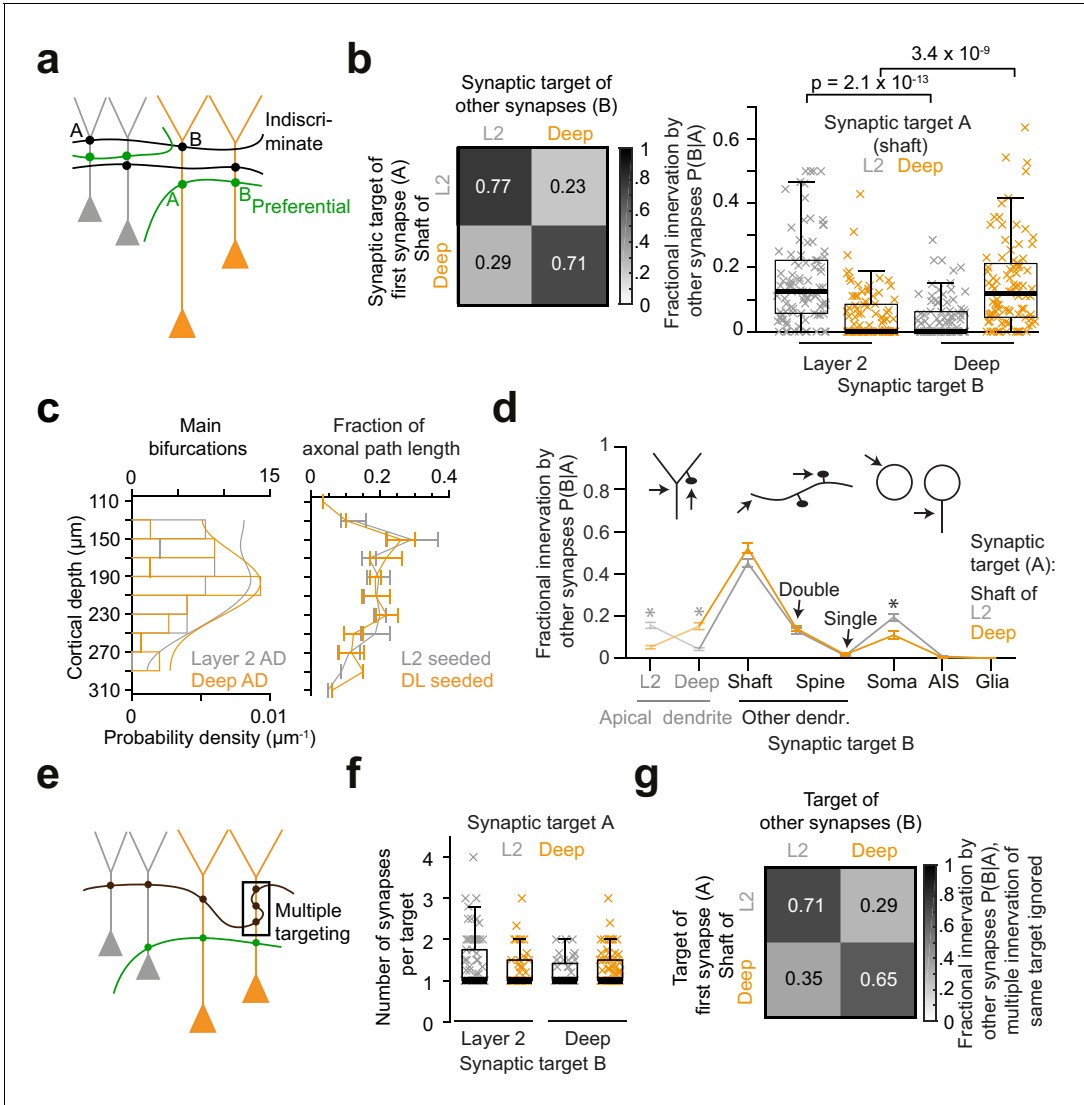

**Figure 3.** Preference of inhibitory axons for the type of postsynaptic pyramidal cell. (**a**) Sketch illustrating two extreme innervation models for AD-targeting inhibitory axons: innervation could be selective for the type of AD (L2 vs L3/5 pyramidal cells, green), or indiscriminate for the type of AD (black). In the latter case, conditional dependence of targeting p(B|A) would be expected to be indistinguishable from overall probability of innervation p(B). (**b**) Conditional dependence of synaptic innervation p(B|A) shown as the mean probability of deep and layer 2 AD targeting (target 'B') given the target of the first synapse of an axon (target 'A', left panel). Probabilities reported as the average fraction of AD synapses onto each AD type. Box plot of AD fractional innervation for axons seeded from layer 2 (n = 92, gray crosses) and deep (n = 91, orange crosses) ADs (right panel), corresponding to the entries in the innervation matrix (left panel). P-values: Wilcoxon rank-sum test. Note that indiscriminate innervation can be revoked. (**c**) Distribution of main bifurcations along cortical depth (n = 41 per AD type; probability density estimates, lines), and distribution of axonal path length along cortical depth (n = 183 axons, of these n = 92 seeded at L2 (gray) and n = 91 seeded at DL ADs (orange)). Note that neither the pre- nor the postsynaptic targets are sorted along the cortical axis, excluding simple layering effects for the conditional innervation (b,c). Error bars indicate mean ± SEM over cortical region (S1, V2, PPC and ACC). (**d**) Mapping of axonal output onto subcellular targets. Error bars indicate mean ± SEM; asterisks: significance of bootstrapping test at p = 0.05 with Bonferroni correction. (**e**) Sketch illustrating innervation of an AD with multiple synapses (box). (**f**) Box plot for comparison of the average number of synapses per individual AD target for axons seeded from L2 (gray crosses) and deep (orange crosses) ADs. Wilcoxon rank-sum test, p = 0.49, 0.23 for L2 and deep ADs, respectively. (**g**) Conditional dependence of AD innervation p(B|A) where multiple innervation of same AD target were ignored (Compare to b). Probabilities indicate fraction of individual ADs targeted by axons seeded from L2 (first row) and deep (second row) ADs.

The online version of this article includes the following source data for figure 3:

**Source data 1.** Conditional innervation probability of sub-cellular structures by AD-targeting putative inhibitory axons (*Figure 3b, d* and *Figure 4b, c*).
**Source data 2.** The coordinate of analyzed AD main bifurcations (y-axis corresponds to the depth relative to pia, *Figure 3c*).
**Source data 3.** Distribution of AD-targeting axonal path length along the cortical depth in S1, V2, PPC and ACC datasets (*Figure 3c*).
**Source data 4.** Average number of synapses on individual AD targets by putative inhibitory axons (*Figure 3f*).
**Source data 5.** The distribution for the number of synapses on individual AD targets by putative inhibitory axons (*Figure 3f, g*).

wanted to determine the variability or consistency of synaptic target preference across these cortical regions. We found remarkable quantitative consistency in innervation preference for axons seeded from L2 ADs (*Figure 4b*). The coefficient of variation (CV) of fractional innervation preference across cortical regions was 19.9%, 39.6%, 19.2% for reinnervation of L2 ADs, DL ADs and somata by L2 AD-seeded axons, respectively. For axons seeded at DL ADs, the CV of fractional innervation preference across cortical regions was 52.3%, 36.6%, 69.1% for reinnervation of L2 ADs, DL ADs and somata, respectively. Only reinnervation of L2 soma by deep-seeded axons showed significant variability across cortical regions (*Figure 4c*, MANOVA test followed by multiple one-way ANOVAs, p = 0.005). Note that all these axons had at least one synapse onto a shaft or spine of ADs, such that axons that innervated no shaft or spine of ADs were not included in this comparison.

## More detailed pyramidal cell type classification

We next wanted to determine whether the observed innervation differences were common to supragranular (L2/3) vs infragranular (L5) pyramidal cells (thus our data was possibly diluted by L3 pyramidal cells classified as 'deep layer'), or whether the putative inhibitory innervation was in fact distinct between L2 and the other pyramidal cells (*Figure 5*). We used two datasets in which directly adjacent to the high-resolution EM data volume a low-resolution EM dataset had been acquired that extended to infragranular layers and allowed the following of ADs to their soma of origin (*Figure 5a*, datasets from PPC and LPtA cortex, *Figure 5—figure supplement 1* for gallery of EM-based pyramidal cell reconstructions). In these datasets we could furthermore distinguish L2 pyramidal cells located directly at the L1-L2 border with a less prominent apical dendrite (L2 marginal neurons, L2-MNs, *Figure 5a*).

In order to also differentiate between L5 pyramidal cells termed 'thick tufted' and 'slender tufted' (*Hübener and Bolz, 1988*; *Hübener et al., 1990*; *Manns et al., 2004*), we used the observation that pyramidal cells with a clear 'slender' morphology (small tuft, few oblique dendrites, [*Larkman and Mason, 1990*]) had smaller soma diameters than those clearly identifiable as thick tufted (*Figure 5b*). In addition, the cortical depth of the main bifurcation (*Staiger et al., 2016*), number of oblique dendrites and diameter of the apical dendrite *Groh et al., 2010*; *Narayanan et al., 2015* have been reported to differ between L5tt and L5st pyramidal cells. We therefore used these four features to cluster L5 pyramidal cells into L5tt and L5st (*Figure 5c*). The resulting subtypes differed

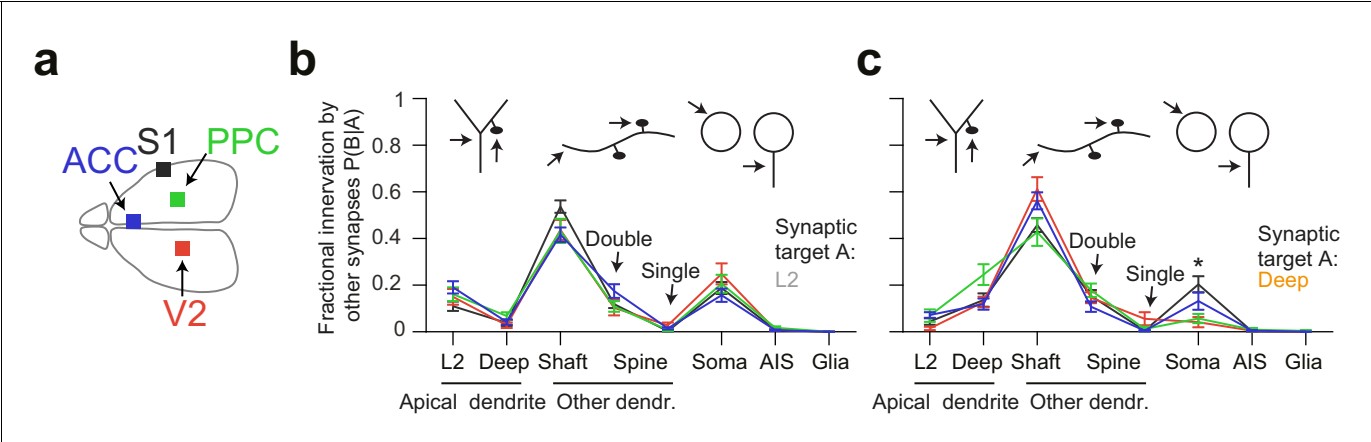

**Figure 4.** Quantitative consistency in the innervation profile of inhibitory axons across four cortical regions. (a) Sketch of the horizontal view of the mouse brain with location of cortical datasets used for comparison. See also *Figure 1a*. (b,c) Comparative analysis across cortical regions. Postsynaptic target specificity for axons seeded from (b) layer 2 ADs (n = 21, 20, 21, 30 for S1, V2, PPC and ACC, respectively) and (c) for axons seeded from deep layer ADs (n = 19, 20, 20, 32 for S1, V2, PPC and ACC, respectively). Note the high level of quantitative consistency of synaptic target fractions across cortices with the exception of somatic innervation in axons seeded from DL ADs. Error bars indicate mean ± SEM. *p < 0.05 MANOVA test followed by multiple One-way ANOVA with Bonferroni correction.

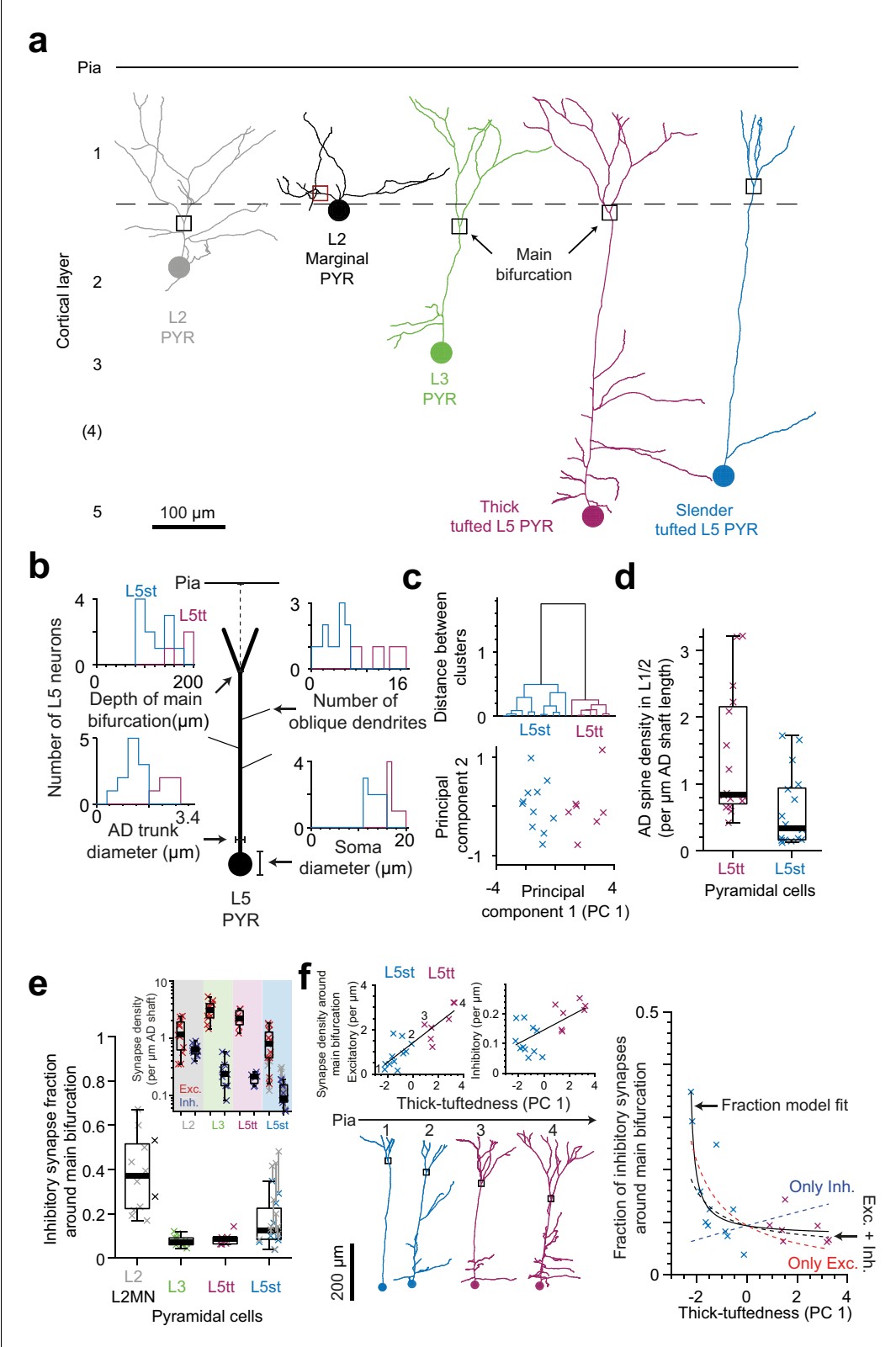

**Figure 5.** Synaptic input composition for L2, L3, and subtypes of L5 pyramidal cells at their main bifurcations. (**a**) Complete skeleton reconstruction of apical dendrites contained within the high- and low-resolution volume of the PPC cortex dataset (PPC-2). Main bifurcation (black/red rectangles) and soma of origin (circles) is marked for L2 (gray, https://wklink.org/7762), L2MN (black, https://wklink.org/5352), L3 (green, https://wklink.org/7652), L5tt (magenta, https://wklink.org/3415) and L5st (blue, https://wklink.org/1435) neurons. (**b**) Sketch illustrating four morphological features of L5 neurons

*Figure 5 continued on next page*

*Figure 5 continued*

used for subtype classification. Insets show histograms of each feature for the L5tt (magenta, n = 7) and L5st (blue, n = 11) neurons. Note the differences between the two cell types. (c) Agglomerative hierarchical linkage tree (upper panel) and scatter plot of the first two principal components (lower panel) for (b). (d) Box plot for comparison of spine density between L5tt (magenta, n = 16) and L5st (blue, n = 17) neurons at their main bifurcation and distal apical tuft. (e) Fraction of putative inhibitory input synapses around the main bifurcation of apical dendrites from layer 2 (gray, n = 10), 2MN (black, n = 2, box plot shared with L2 group), 3 (green, n = 10), 5tt (magenta, n = 7), 5st (blue, n = 11) pyramidal cells (box plots) and density of excitatory (red crosses) and putative inhibitory (blue crosses) synapses at the main bifurcation (inset). Note clear distinction of synaptic input composition for L2 vs L3 and L5tt pyramidal cells at their main bifurcation. Kruskal-Wallis test, $p < 10^{-4}$. (f) Scatter plot of synapse densities (left panels) and putative inhibitory fraction (right panel) of L5 neurons in relation to their thick-tuftedness (PC 1). Skeleton reconstruction of neurons indicated by numbers in the left panel demonstrate the relationship of PC 1 to AD tuft size (https://wklink.org/1293). Excitatory synapse density and putative inhibitory fraction have linear ($E_{density} = 0.44 \times PC1 + 1.4$, $R^2 = 0.74$, $p < 10^{-5}$) and non-linear relationships ($I_{fraction} = \frac{0.043 \times PC1 + 0.13}{0.58 \times PC1 + 1.40}$, $R^2 = 0.70$, $p < 10^{-7}$) to thick-tuftedness, respectively. Dashed lines in right panel indicate models using only linear excitatory (red), linear putative inhibitory (blue) or combined excitatory and putative inhibitory linear relationships (black).

The online version of this article includes the following source data and figure supplement(s) for figure 5:

**Source data 1.** Morphological properties of L5 pyramidal neurons (*Figure 5b, c, f*).
**Source data 2.** Dendritic spine density for subtypes of L5 pyramidal neurons (*Figure 5d*).
**Source data 3.** More detailed cell-type comparison of the synapse density and fraction at the main bifurcation of ADs (*Figure 5e*).
**Figure supplement 1.** Skeleton reconstruction of apical dendrites of pyramidal neurons in PPC and LPtA cortical regions.

in spine density (*Figure 5d*, 1.4 ± 0.24 vs. 0.59 ± 0.13 spines per µm path length for L5tt and L5st, respectively, Wilcoxon rank-sum test, p = 0.004).

When we then analyzed the fractional putative inhibitory synapse densities at the main bifurcation for L2, L3, L5tt and L5st pyramidal cells (*Figure 5e*), we found that, in fact, L2 pyramidal cells were unique in their high putative inhibitory synapse fraction. L3 pyramidal cells, to the contrary, showed the lowest putative inhibitory synapse fraction, comparable to L5tt pyramidal cells (putative inhibitory synapse fractions of 38.0 ± 4.9% vs 7.8 ± 0.8% vs 8.7 ± 1.1% vs 15.3 ± 3.0% for L2, L3, L5tt and L5st pyramidal cells, respectively, mean ± SEM, n = 40, Kruskal-Wallis test, $p < 10^{-4}$, Post-hoc Tukey's range test, $p = 2.7 \times 10^{-5}$ and 0.002 for L2 vs. L3 and L5tt, respectively).

The putative inhibitory synapse fraction in L5 pyramidal cells was strongly anti-correlated to the 'thick-tuftedness' of the L5 pyramidal cell (quantified as the first principal component in the four-parameter space used for L5tt vs L5st distinction above, *Figure 5b,c,f*). Interestingly, this effect was present although we had found that the shaft of L5st apical dendrites had substantial input from spine-preferring axons (*Figure 2b–c*, Table 4 in *Supplementary file 1*, 44% and 60% of shaft synapses onto L5st ADs were likely excitatory in L1 and L2, respectively), and we had corrected the synapse densities for L5st neurons accordingly (*Figure 5e*). The positive correlation between excitatory synapse density and the thick-tuftedness of L5 neurons was the main driver for the reduction in putative inhibitory fraction in L5tt neurons (*Figure 5f*, left panel, Linear fit: $R^2 = 0.74$, $p < 10^{-5}$). Using only the linear fit to excitatory synapse density and the average putative inhibitory synapse density, 55% variability in the putative inhibitory fraction of L5 neurons could be explained (*Figure 5f*, right panel, red dashed line). For comparison, a linear fraction model fitted to the data explained an additional 15% of variance in the putative inhibitory fraction (*Figure 5f*, right panel, black solid line, $R^2 = 0.70$, $p < 10^{-7}$, Linear fraction model, n = 4 parameters).

Two identified L2 marginal neurons (L2MN, with an AD that runs oblique to the pial surface, *Figure 5a*, [*Larkman and Mason, 1990*; *Luo et al., 2017*]) showed putative inhibitory synapse fractions at their main bifurcation that were consistent with that of other L2 neurons (*Figure 5e*, 53.2% and 27.8% putative inhibitory fraction for L2MN).

## Effect of pyramidal cell type vs. distance to soma

The observed pyramidal-cell-type specific putative inhibitory innervation was based on a cell type classification that relied largely on the cortical layer position of the pyramidal cell's soma (*Figure 5a*). Since main bifurcations were located approximately at the same cortical depth around the L1-to-L2 border (*Figure 5a*), the cell types differed substantially in their dendritic distance between the main bifurcation and the cell body (i.e. the length of their apical dendrite trunks). We therefore analyzed the relation between the fraction of putative inhibitory synapses at the main bifurcation and the distance of the main bifurcation to the soma and found a very strong distance-

dependence (*Figure 6a,c*, $R^2$ = 0.73 for single exponential fit, n = 81). This dependence was even evident within the population of L2 pyramidal cells, consistent with a possible synaptic specificity mechanism that is dependent on the distance to the cell body. Interestingly, this dependence was weaker for the absolute density of excitatory or putative inhibitory synapses, compared to their ratios (*Figure 6b*, $R^2$ = 0.01, 0.56, respectively). The relationship was consistent for all pyramidal cell types except L5st (*Figure 6a*). The distance-dependence of the putative inhibitory input fraction was also found for apical dendrite locations before and after the main bifurcation within the L2 pyramidal cell population (*Figure 6c*).

### Putative inhibitory synapse fractions at the distal apical tuft

Finally, we analyzed the putative inhibitory and excitatory innervation of ADs towards their upper ends in L1 (*Figure 7a,b*, LPtA dataset). This was of special interest since the density of dendritic spines in hippocampal pyramidal cells had been previously found to decrease by about 5-fold towards the AD's tip (*Bannister and Larkman, 1995*; *Megías et al., 2001*). We wanted to understand whether this was similar in cortical pyramidal cells, and whether this was accompanied by a drop (or increase) in putative inhibitory synapse density.

We mapped all input synapses onto 14 additional apical tuft dendrites (*Figure 7a*, n = 4506 synapses) and found that, first, the excitatory synapse density of L5tt pyramidal cells decreased 3-fold towards their distal tufts. This was not accompanied by a drop in putative inhibitory inputs, which remained almost constant in density (*Figures 5e* and *7b*, *Figure 7—figure supplement 1a–b*). Thus, L5tt pyramidal cells received enhanced putative inhibitory input at their distal dendrites (*Figure 7b*, putative inhibitory input fraction 22.5 ± 1.7% at the distal tips, mean ± SEM, n = 16 dendritic segments, p < $10^{-3}$, Wilcoxon rank-sum test compared to MB). For L5st pyramidal cells, the putative inhibitory input fraction at their distal apical tufts was also higher than their main bifurcations, with about a third of their synapses putative inhibitory (*Figure 7b–c*, 15.3 ± 3% and 31.3 ± 1.8% putative inhibitory synapse fraction at main bifurcation and distal tips of L5st, respectively, mean ± SEM, n = 17, Wilcoxon rank-sum test, p = 0.005). Together, this demonstrated increased inhibition for all L5 pyramidal neurons at their distal AD.

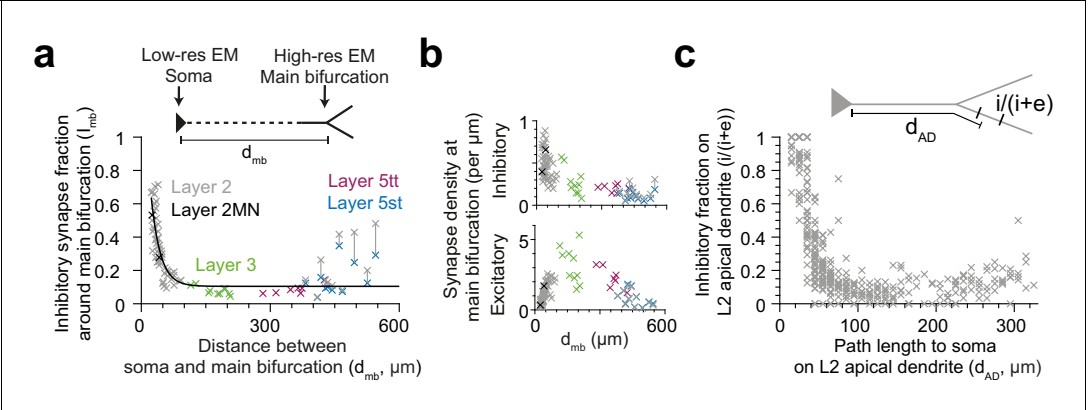

**Figure 6.** Dependence of inhibitory input fraction at the main bifurcation of apical dendrites to the distance to soma. (a) Relationship between distance of main bifurcation to soma and putative inhibitory fraction at the main bifurcation for L2 (n = 51, gray), L2MN (n = 2, black), L3 (n = 10, green), L5tt (n = 7, magenta) and L5st (n = 11, blue) ADs. Black line indicates single exponential regression ($I_{mb} = 1.57 * e^{-0.047 * d_{MB}} + 0.1$, $R^2$ = 0.73, p < $10^{-22}$). (b) Same as in (a) for putative inhibitory and excitatory synapse densities, respectively. Note tight distance dependence for the putative inhibitory input fraction (a), but not the absolute synapse densities (b). (c) Relationship between path distance to soma on the apical dendrite and the fraction of putative inhibitory synapses on L2 apical dendrites (n = 66, S1, V2, PPC, ACC, PPC-2 and LPtA datasets, n = 12,532 synapses). Each cross represents the putative inhibitory fraction (i/(i+e)) for a single apical dendrite within a 10 µm path length range ($d_{AD}$) to cell body of origin (inset).
The online version of this article includes the following source data for figure 6:

**Source data 1.** Path distance between soma and main bifurcation (MB) with the corresponding synapse densities and fractions at MB (*Figure 6a, b*).
**Source data 2.** The fraction of putative inhibitory synapses along the apical dendrite of L2 neurons (*Figure 6c*).

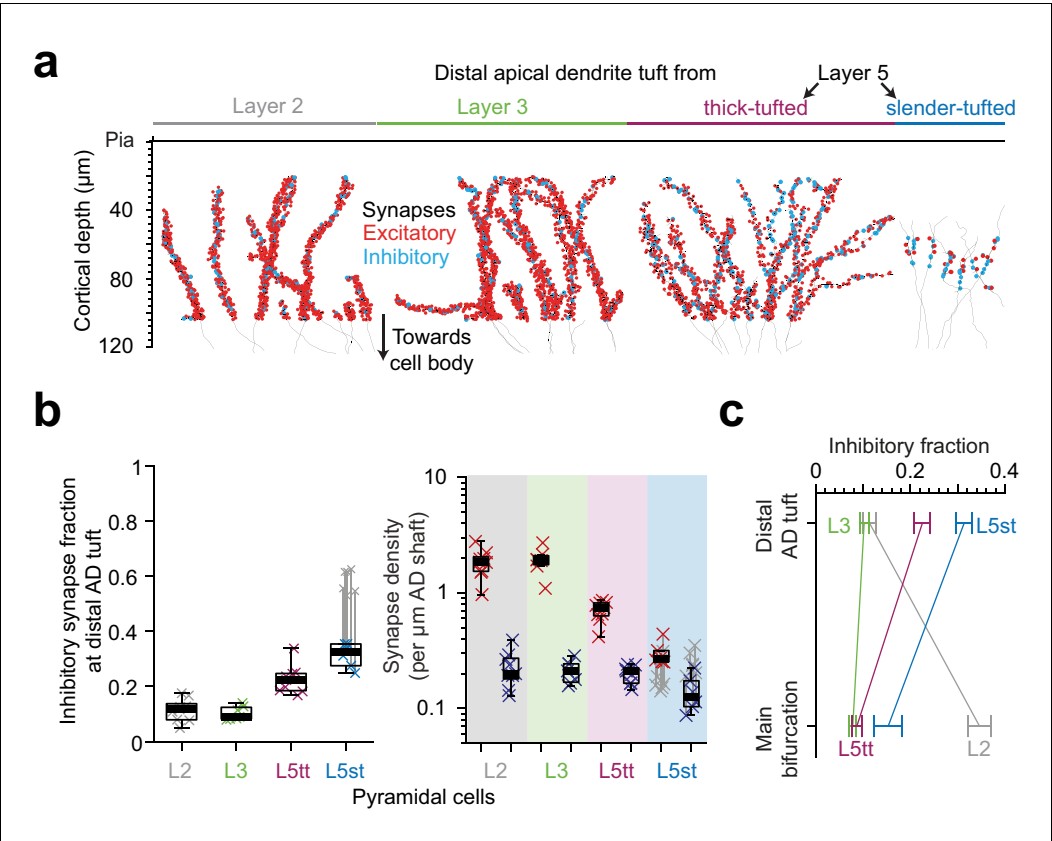

**Figure 7.** Inhibitory input fraction at distal apical tuft dendrites. (a) Skeleton reconstruction of 14 L2, L3, L5tt and L5st ADs with all synapses mapped within the high-resolution EM image volume in layer 1 of LPtA (L2, L3, L5tt, https://wklink.org/9047) and PPC (L5st, https://wklink.org/8828) cortex (red: excitatory synapses, n = 3,812, cyan: putative inhibitory synapses n = 694). Note the difference in synapse densities (red/cyan sphere density) for layer 2, 3, 5tt, 5st ADs. (b) Box plot for fraction of putative inhibitory synapses at the distal apical dendritic site from L2 (n = 9 individual branches, gray crosses), L3 (n = 7, green crosses), L5tt (n = 9, magenta crosses) and L5st (n = 6, blue crosses) pyramidal cells and box plot for density of excitatory (red crosses) and putative inhibitory (blue crosses) synapses at the distal tuft (right panel). Note the clear distinction of synaptic input composition in the distal tuft for L2 vs. L3 vs. L5tt vs. L5st pyramidal cells (Kruskal-Wallis test, p < $10^{-4}$) mainly due to differences in excitatory synapse density. (c) Summary of distinct putative inhibitory input fraction at the main bifurcation and distal tufts of apical dendrites for four main classes of pyramidal cells in the cerebral cortex. Note that only L3 pyramidal cells show approximately homogeneous ratio of putative inhibitory and excitatory synaptic inputs along their apical dendrites. Error bars indicate mean ± SEM.

The online version of this article includes the following source data and figure supplement(s) for figure 7:

**Source data 1.** The synapse density and fraction in the distal part of apical dendrites (*Figure 7b*).

**Figure supplement 1.** Synaptic composition for layer 2, 3, 5tt and 5st apical dendrites (ADs) along the upper cortex.

---

For L2 pyramidal cells, to the contrary, putative inhibitory synapse density decreases ~2 fold towards the distal tuft (*Figures 5e* and *7b*, 0.47 ± 0.02 vs 0.23 ± 0.03 putative inhibitory synapses per μm dendritic path, mean ± SEM, n = 62, Wilcoxon rank-sum test, p < $10^{-4}$) while the excitatory input increases by ~1.5 fold, yielding an about 3-fold reduced putative inhibitory input fraction towards their distal tufts (*Figures 5e* and *7b*, 34.6 ± 2.5% putative inhibitory input fraction at main bifurcation vs 11.4 ± 1.4% at distal tufts, Wilcoxon rank-sum test, p < $10^{-4}$).

Finally, we found that L3 pyramidal cells show no substantial dependence of their synaptic input fractions on the position along the apical dendrite: they showed low putative inhibitory input fraction around their main bifurcation (*Figure 7b–c*, 7.8 ± 0.8%, mean ± SEM, n = 10, similar to the L5tt cells,

Wilcoxon rank-sum test, p = 0.36), which only slightly increased towards their distal tufts (10.2 ± 0.9%, mean ± SEM, n = 7 tuft branches, similar to L2 cells, Wilcoxon rank-sum test, p = 0.84).

With this, we find that the pyramidal cells classified as L2, L3, L5tt and L5st show each distinct putative inhibitory innervation profiles along their apical dendrites, providing succinct possibilities for putative inhibitory control of excitatory inputs in the cortex (*Figure 7c*, *Figure 7—figure supplement 1a–c*) and emphasizing the need to consider L2 and L3 pyramidal cells (and subtypes of L5 pyramidal cells) as separate entities.

## Discussion

We obtained a quantitative connectomic input map of pyramidal cell apical dendrites in several areas of the mouse cerebral cortex. We found distinct profiles of putative inhibitory to excitatory synaptic innervation ratio for all types of pyramidal cells (*Figure 7*): L2 pyramidal cells received substantial inhibition at their apical dendrites' main bifurcation, which decreased about 3-fold towards the tips of the apical dendrites; in thick-tufted layer 5 pyramidal cells, this profile was inverted with a decreased fraction of putative inhibitory inputs at the main bifurcation. L5 slender-tufted cells followed the L5tt profile but received about 50% higher putative inhibitory input fraction overall. L3 pyramidal cells were the only cell type to show a homogeneous putative inhibitory input across upper cortex of about 10%. At the level of axons, putative inhibitory innervation exhibited connectomic specificity with axons preferring the innervation of either L2 or deeper-layer pyramidal cells (*Figure 3*). These data indicate a unique innervation profile for each pyramidal cell type, and point to a high variance of the cell-type specific ratio of inhibition and excitation in the cortex. Our findings were remarkably consistent for the investigated cortices (S1, V2, PPC and ACC, *Figure 4*).

### Distinction of excitatory vs inhibitory synapses

For the distinction of excitatory vs. inhibitory synapses in single high-resolution EM images, the symmetry of pre- and postsynaptic densities and the properties of the synaptic cleft have been developed as criteria (*Gray, 1959*; *Colonnier, 1968*). In large-scale 3D EM data, it is possible to image and analyze extended stretches of axons and assign their likely inhibitory vs. excitatory characteristic based on the target distribution of their synapses. This is especially fruitful when for the acquisition of large image volumes, compromises in in-plane image resolution have to be made as is the case for SBEM (insets in *Figure 1c*; see also [*Staffler et al., 2017*]). To calibrate the target bias of axons for innervating shafts vs. spines of dendrites, we mapped the preference of AD-targeting axons to innervate dendritic shaft and spines (*Figure 2a–c*), and found in all but one case clearly distinct axonal populations with almost exclusive targeting of shaft vs. spines. These axonal preferences appear even more distinct than the axonal preferences recently reported in L4 of mouse S1 cortex (*Motta et al., 2019*).

### Synaptic vs. non-synaptic inhibition

In physiological experiments, evidence for a strong effect of GABA on regenerative activity in apical dendrites of thick-tufted L5 cells via non-synaptic GABA-B receptors had been reported (*Pérez-Garci et al., 2006*; *Oláh et al., 2009*; *Pérez-Garci et al., 2013*; *Abs et al., 2018*). Together with our evidence on low rates of synaptic putative inhibitory innervation at the main bifurcation of these pyramidal cells, this may indicate that L5tt pyramidal cells are controlled by overall and slow inhibition at their main bifurcations, but specific synaptic innervation at their distal dendrites, while L2 pyramidal cells receive specific synaptic inhibition at their main AD bifurcations. Whether L3 pyramidal cells share the electrical properties of L5 pyramidal cells is still controversial (*de Kock and Sakmann, 2009*; *Barth and Poulet, 2012*). Their synaptic input composition at the main bifurcation as shown here would indicate commonality; but this could be counteracted by different distribution of active transmembrane conductances (*Waters et al., 2003*; *Ledergerber and Larkum, 2012*). Their distinct putative inhibitory input profile towards the tips of the apical dendrites may suggest that they allow for specific computations not shared by L2 or L5 pyramidal cells.

### Distal inhibitory domains

Spatial distribution of excitatory and inhibitory synapses on the dendritic surface has substantial effects for the integrative properties and output of neurons (*Rall, 1959*; *Polsky et al., 2004*;

*Katz et al., 2009*). Previous studies in hippocampal pyramidal cells had reported a substantial increase in inhibitory synapse fraction towards the distal tuft dendrites which is driven mainly by a drop in excitatory synapse density (*Bannister and Larkman, 1995*; *Megías et al., 2001*; *Bloss et al., 2016*). We find that L5 pyramidal cells in cortex exhibit a similar distal putative inhibitory input domain. Theoretical work indicated that this is potentially a very powerful inhibitory effect (*Gidon and Segev, 2012*). This distal inhibitory domain, however, is absent in the distal tufts of L2 and L3 pyramidal neurons.

### Distinction of L2 vs L3 pyramidal cells in cortex

Importantly, our data shows a striking specialization of L2 vs L3 pyramidal cells, which has so far evaded analysis (for example, [*Iascone et al., 2018*]). Only L3 (and some degree L5st), but not L2 or L5tt pyramidal cells, keep a roughly even putative inhibitory to excitatory synaptic input ratio. Recent work using intracellular Gephyrin tagging for the detection of inhibitory synapses and counting of spines for excitatory synapses in LM (*Iascone et al., 2018*) underestimated the rate of spines by about ~20% compared to our complete 3D EM based sampling, resulting in an overestimation of the inhibitory-to-excitatory input fraction for L3 pyramidal cells (see also [*Chen et al., 2012*]). Our results emphasize the importance of separately analyzing L2 and L3 (and subtypes of L5) pyramidal cells in cortex (*Figures 1*, *5* and *7*, see also [*Meyer et al., 2011*]).

### Conclusion

In conclusion, our findings describe non-random connectivity at the apical tuft input domain of pyramidal neurons in cortex. They imply unique innervation patterns for the apical dendrites of L2 vs L3 vs L5 pyramidal cells yielding highly variable and cell-type specific putative inhibitory-to-excitatory synaptic input compositions, and suggest differential modes of inhibitory operation for the main classes of pyramidal cells in the cortex.

## Materials and methods

### Animal experiments

All experimental procedures were performed according to the law of animal experimentation issued by the German Federal Government under the supervision of local ethics committees and according to the guidelines of the Max Planck Society. The experimental procedures were approved by Regierungspräsidium Darmstadt, under protocol ID V54 - 19c20/15 F126/1015 (LPtA, PPC-2) or V54 – 19 c 20/15 – F126/1002 (V2, PPC, ACC). The S1 sample was prepared following experimental procedures approved by Regierung von Oberbayern, 55.2-1-54-2532.3-103-12.

### S1, LPtA sample preparation

The cortical tissue processing, 3D electron microscopy and data alignment for S1 and LPtA samples were described previously in *Berning et al. (2015)* and *Drawitsch et al. (2018)*, respectively. V2, PPC, ACC and PPC-2 samples were processed as follows.

### Transcardial perfusion

Adult (P56 – 57) wild type mice (C57BL/6J) were injected with general analgesia (0.1 mg/kg buprenorphine (Buprenovet, Recipharm, France) and 100 mg/kg Metamizol (Metamizol WDT, WDT, Germany)). Next, animals were anesthetized by inhalation of isoflurane (3–3.5% in carbogen) and were perfused transcardially using 15 ml of cacodylate buffer (150 mM, Serva, Heidelberg, Germany, pH = 7.4) followed by 30 ml of fixative solution at a flow rate of 10 ml/min. The fixative solution was 2.5% PFA (Sigma-Aldrich, Germany), 1.25% glutaraldehyde (Serva) and 0.5% $CaCl_2$ (Sigma-Aldrich) in 80 mM cacodylate buffer (pH = 7.4). The animal was decapitated and the skull was removed to expose the brain. The head was next submerged in fixative solution overnight at 4°C.

### Cortical region targeting and tissue extraction

First, a 600 µm coronal slice containing the region of interest was acquired using a vibrating microtome (Microm HM 650V, Thermo Fisher Scientific, USA) guided by a reference atlas (*Franklin and Paxinos, 2008*). Next, we used a biopsy punch (1 mm in diameter, KAI medicals, USA) to extract a

cylindrical piece of cortex containing layers 1–3 (*Figure 1—figure supplement 1*). This tissue was then incubated for 3–4 hr (V2, PPC and ACC) or overnight (PPC-2) in cacodylate buffer. The PPC-2 sample was also subjected to confocal laser scanning light microscopy as previously described (*Drawitsch et al., 2018*).

## En-bloc sample preparation for 3D electron microscopy

Samples were prepared for serial block-face electron microscopy following a slightly modified en-bloc staining method as described in *Hua et al. (2015)*. In short, cortical tissue was rinsed in cacodylate buffer for 30 min before any staining material was applied. Next, it was transferred into 2% $OsO_4$ (Serva, Germany) in cacodylate buffer for 90 min. The sample was then treated with 2.5% ferrocyanide (Potassium hexacyanoferrate trihydrate, Sigma-Aldrich, Germany) in cacodylate buffer for 90 min and 2% buffered $OsO_4$ for 45 min. We then rinsed the tissue in cacodylate and ultrapure water (Biochrom, Germany) for 30 min each. Osmium content of the sample was amplified by treatment with saturated aqueous thiocarbohydrazide (TCH, Sigma-Aldrich, Germany) and 2% aqueous $OsO_4$ for 60 and 90 min, respectively. The sample was moved to 2% uranyl acetate (Serva) solution for overnight incubation at 4°C. The following day, the tissue, still in uranyl acetate, was warmed to 50°C for 120 min in an oven (Memmert, Germany). This was followed by incubation in lead aspartate at 50°C for 120 min. The lead aspartate solution was prepared by dissolving 0.066 g lead nitrate (Sigma-Aldrich, Germany) in a 10 ml 0.03 M aspartic acid (Serva, Germany) solution and adjusting the pH to 5.0.

Next, the cortical tissue was dehydrated by incubation in 50–100% ethanol gradient (Serva, Germany). This was followed by at least three 20–45 min incubation steps in pure acetone (Serva, Germany). The sample was then transferred to a 1:1 mixture of acetone and Spurr's resin (4.1 g ERL 4221, 0.95 g DER 736 and 5.9 g NSA, 113 µl DMAE, Sigma-Aldrich, Germany) for 3–4 hr with slow/no rotation. Tubes were opened to allow for acetone evaporation overnight (V2, PPC and ACC samples). At this stage, the PPC-2 sample was transferred to a 3:1 mixture of Spurr's resin and acetone instead. The following day, the infiltration process continued for two 3 hr (PPC-2) or one 6 hr (V2, PPC, ACC) incubation steps in pure Spurr's resin mixture. The tissue was then transferred to a flat-embedding mold and cured at 70°C for at least 48 hr. Specific time and temperature of dehydration and embedding steps are detailed in Table 2, *Supplementary file 1*.

Note that all procedures were performed at room temperature in 2 ml reaction tubes (Eppendorf, Germany) unless stated otherwise. In addition, the initial incubation steps (until dehydration) were performed with the aid of an automatic microwave tissue processor (Leica EM AMW, Leica, Germany) for V2, PPC, and ACC samples. Finally, treatment steps were interleaved with two 30 min washing steps in ultrapure water (from TCH step until dehydration initiation).

## Serial block-face electron microscopy (SBEM)

The samples were excised from the resin block and mounted on an aluminum pin using epoxy glue (Uhu plus schnellfest, Uhu, Germany). They were then trimmed to a block-face area of ~750 µm x 750 µm using a diamond head trimming machine (Leica EM TRIM2, Leica, Germany). In addition, tissue was sputter coated with 100–200 nm of gold (Leica ACE600 Sputter Coater, Leica, Germany) to increase conductivity and reduce charging artefacts.

The SBEM microtome (courtesy of W. Denk) was fit inside the door of a scanning electron microscope (Table 1 in *Supplementary file 1*, FEI, Thermo Fisher Scientific, USA). The microtome and microscope were controlled using custom written software during the volume imaging process; focus and stigmation were corrected manually (V2, PPC, ACC and LPtA) or using custom written auto-correction routines (S1, PPC-2). The region of interest was imaged using overlapping image tiles and cutting direction was along the tangential (S1, V2, PPC, ACC, PPC-2) or radial (LPtA) axes of cortex. The region of interest was targeted to an area close to layer 1/2 border (S1, V2, PPC and ACC, *Figure 1a*). The final imaged volume and the nominal voxel size is detailed in Table 1, *Supplementary file 1*. Note that the LPtA and PPC-2 datasets were adjacent to datasets extending to the middle of layer 5 imaged at lower resolution ($22.48 \times 22.48 \times 30$ and $44.96 \times 44.96 \times 120$ $nm^3$ for PPC-2 and LPtA, respectively).

### Image alignment

The alignment for PPC, V2 and ACC dataset was done using custom-written MATLAB (Mathworks, USA) routines. We used cross-correlation or speeded-up robust feature (SURF) detection in the overlap region to measure the relative shift between image patches. These patches were full image tiles (V2, ACC) or tile sub regions (PPC, 256 × 256 pixels). The position of each patch was then globally optimized using least-square regression. The image volume was then partitioned into 1024 × 1024 × 1024 voxel blocks and written into webKnossos (*Boergens et al., 2017*) three-dimensional format. The dataset was then transferred to the data store accessible by webKnossos for in-browser neurite skeleton reconstruction and synapse annotation.

The PPC-2 dataset was aligned using the affine alignment method from *Scheffer et al. (2013)*. The routines were modified to give sub regions of image tiles unique affine transformations. In addition, methods to exclude featureless blood vessel and nuclei were improved.

### Reconstruction and synapse annotation analysis

The skeleton reconstruction and synapse annotations were saved as a NML (XML-based format) file. They were subsequently parsed into a MATLAB (release 2014b-2019a) class with node and edge list properties using custom-written C++ routines. Each node had accompanying attributes, such as, comment string and coordinate. These attributes were used to extract different features of the annotation as described in the following sections.

### Apical dendrite (AD) definition and classification

Apical dendrites (ADs) were identified based on their radial direction and diameter (~1–3 μm). The soma morphology and axon initial segment direction (towards white matter (WM)) of candidate pyramidal cells were examined where possible.

Within the S1, V2, PPC and ACC datasets, ADs were classified depending on the existence of soma in the image volume (*Figure 1a*). ADs with soma in the image volume were classified as layer 2 (L2) and other apical dendrite were classified as deep layer (DL or L3/5) ADs.

The depth of the apical dendrite's source soma relative to pial surface was used to differentiate layer 2, 3 and 5 cells in the LPtA and PPC-2 datasets that were contained in layers 2–5 (Table 5 in *Supplementary file 1*). Note that, LPtA and PPC cortical regions do not possess a prominent layer 4 (*Kolb and Walkey, 1987*).

The difference between nominal and actual cutting thickness in the LPtA dataset was resulting in apparently ellipsoid somata (compressed along the cutting axis). To obtain an accurate estimate of soma depth relative to pial surface, the section thickness was corrected (by a factor of 1.49), assuming soma dimensions to be similar in-plane and along the cutting direction.

The main bifurcation of apical dendrites was defined as the branching point with two daughter branches of similar thickness and branching angle (resulting in a 'Y' shape, *Figure 1c*).

We used the ACC dataset to reconstruct all ADs (*Figure 1b*, blue-green). They were detected by examining the border of dataset facing WM for deeper layer ADs (n = 152) and by identifying all L2 pyramidal neurons contained for L2 ADs (n = 61). The AD locations were used as seed points for manual annotation ignoring spines.

### Identification of layer 2 marginal neurons (L2MN) and slender-tufted L5 (L5st) ADs

The AD morphology was reconstructed for all ADs investigated to identify the type of pyramidal neuron it belongs to (*Figure 5a*, *Figure 5—figure supplement 1a*). Next, we set out to find the less frequent subtypes for L2 and 5 neurons.

We used the orientation of the apical dendrite to find layer 2 marginal neurons that have an oblique AD which runs parallel to pial surface (*Larkman and Mason, 1990*; *Luo et al., 2017*; *Figure 5a*, *Figure 5—figure supplement 1a*, black reconstructions).

We also found apical dendrites of L5st neurons (*Figure 5a*, *Figure 5—figure supplement 1a*, blue reconstructions) based on the difference in their soma diameter (*Larkman and Mason, 1990*). We also measured their AD diameter, number of oblique dendrites and depth of main bifurcation to perform a hierarchical clustering. Using the correlation of the actual distance between clusters and the height of their link in the hierarchical tree, we found the average cosine distance to be the most

consistent metric (one minus cosine of the angle between points in the four-dimensional morphological feature space). We found two clusters containing L5tt (n = 7) and L5st (n = 11) neurons (*Figure 5b,c*). Furthermore, we performed principal component analysis of the L5 morphological features. Since the first principle component (PC 1) correlated with the thick-tuftedness of L5 neurons, we investigated the linear relationship between PC 1 and synapse densities. Additionally, we used the fitted linear models to synapse density to predict the putative inhibitory fraction (*Figure 5f*, right panel, dashed lines). For comparison, we fitted a non-linear model (fraction of two linear expressions) to the putative inhibitory fraction (*Figure 5f*, right panel, solid line). We also compared the spine density between L5tt and L5st neurons at their main bifurcation and distal apical tuft (*Figure 5d*).

We approximated the soma volume using an ellipsoid by measuring the diameter along the three Cartesian dimensions of the dataset. We then calculated the diameter of the sphere that has an equivalent volume. To measure AD diameter, we determined the diameter of the sphere that has an equivalent area to the ellipse approximation of the AD trunk cross-section about 50 μm from the soma.

## Complete synaptic input mapping of apical dendrites

Apical dendrites and their associated spines were skeleton reconstructed and the synapses on their shaft and spines were annotated either within a bounding box of size 20 × 20 × 20 μm$^3$ usually around the main bifurcation (*Figures 1c–g*, *5e*, *6a* and *7a–b* (L5st group), n = 20 for S1, V2, PPC, n = 40 for PPC-2, n = 22 for ACC) or throughout the dataset (*Figure 7a* (except for the L5st group), *Figure 7—figure supplement 1a*, n = 11 for LPtA, n = 6 for ACC, n = 4 for V2, PPC and ACC) by a neuroscientist expert annotator (JO or AK).

Synapses were identified within the SBEM data based on the presence of vesicle cloud and postsynaptic density as described previously (*Schmidt et al., 2017*). We also annotated spine neck synapses and double innervations of spine heads by two axonal boutons (*Figure 2e*; *Kubota et al., 2007*). Shaft synapses, synapses on spine necks and secondary spine innervations were counted as putative inhibitory synapses and primary spine innervations were counted as excitatory synapses (*Figures 1*, *5* and *7*) with the exception of L5st neurons (*Figures 5e* and *7b*, uncorrected measurements in gray, Also see section: Identity estimation for shaft and spine synapses). All synapse annotations were validated by an additional expert annotator, and only synapses where annotators agreed were used for analysis.

The fraction of putative inhibitory synapses was defined as the number of putative inhibitory synapses divided by the total number of synapses (*Figure 1f,g*, *Figures 5e* and *7b*). The putative inhibitory and excitatory synapse density was calculated by dividing the number of synapses by the path length of the apical dendrite shaft (*Figure 1d,e*, insets in *Figures 5e* and *7b*). Shaft path length was measured by removing the spine necks from the skeleton reconstructions and summing the lengths of the remaining edges. Note that putative inhibitory input was measured separately for each distal tuft branch in the LPtA dataset resulting in 9, 7, 9 and 6 dendritic segments for L2, 3, 5tt and 5st cells, respectively (*Figure 7a,b*).

## Apical dendrite diameter and synapse density per unit surface area

Apical dendrite diameter was measured every ~2–3 μm along the path of each dendrite to get an average diameter (*Figure 2f*). The average diameter in combination with the path length of the dendrite was used to calculate the surface area of the dendrite (AD surface = π x path length x average diameter). We then calculated the synapse density per unit surface of the dendrite by dividing each synapse count by the surface area of the AD (*Figure 2f–g*).

## Inhibitory input fraction mapping in upper cortex

We first estimated the approximate location of each dataset relative to pia (*Figure 1a*, 125, 215, 170, 110, 10 and 20 μm for S1, V2, PPC, ACC, PPC-2 and LPtA, respectively) based on their position in coronal overview images and transformed all reconstructions into a common coordinate system (*Figure 7—figure supplement 1a*). Next, we partitioned these reconstructions (n = 141, total AD shaft path length = 14.1 mm) into virtual 100 μm thick cortical tangential sections. This process resulted in dendritic segments at each depth bin. We then combined synapse counts and path

lengths from all segments within each virtual cortical section to calculate the total putative inhibitory fraction and synapse densities (lines in *Figure 7—figure supplement 1b–c*). We also used the 95% bootstrap confidence interval to estimate the dependence of the total average on the sample composition (shades in *Figure 7—figure supplement 1b–c*, n = 10,000 bootstrap samples).

### Soma, main bifurcation distance effect on synapse composition at the main bifurcation

We annotated the apical trunk connecting the soma to main bifurcation within the high- and low-res EM data volumes for the L2 (n = 51, S1, V2, PPC, ACC, PPC-2), L2MN (n = 2, PPC-2), L3 (n = 10, PPC-2), L5tt (n = 7, PPC-2), L5st (n = 11, PPC-2) pyramidal cells and considered the path length of these annotations as the soma to main bifurcation distance (*Figure 6a–b*).

### Inhibitory fraction along the AD of L2 pyramidal neurons

We binned the excitatory and putative inhibitory synapses based on their path distance to soma in the L2 pyramidal ADs (*Figure 6c*, n = 66 L2 dendritic segments, n = 12,532 synapses, bin size = 10 µm). This allowed us to measure the fraction of putative inhibitory synapses as a function of dendritic path distance to the soma (10–330 µm distance range). Soma distance bins with less than four synapses were merged to their immediate neighbor that contained at least four synapses. This was to avoid extreme values introduced by computing the putative inhibitory ratio in a bin with low synapse count.

### Identity estimation for shaft and spine synapses

A random subset of axons targeting shaft or spine of ADs were reconstructed (*Figure 2a–c*, n = 142 and 288 for datasets in layers 1 and 2, respectively). The postsynaptic targets were identified for at least six additional targets of each axon if possible. Next, we determined whether the postsynaptic target was single-innervated spine or shaft/non-spine (this included shaft, spine neck, double-innervated spine and soma). We then plotted the histogram (bin size = 0.1) and the probability density estimate (bandwidth = 0.0573) of single-innervated spine innervation fraction for axons seeded from shaft or spine of ADs within layers 1 and 2 (*Figure 2b*).

We then classified axons into excitatory and putative inhibitory by using a 50% threshold on the single-innervated spine innervation fraction (*Figure 2b*, dashed lines). Axons with majority of their synapses onto single-innervated spines were counted as excitatory and the putative inhibitory axons had less than half of their synapses on single-innervated spines. Next, we calculated the fraction of excitatory input on each structure (Table 4 in *Supplementary file 1*). Subsequently, we corrected synapse densities and fraction of putative inhibitory synapses based on the excitatory synapse fraction on each structure. We then compared the synaptic measurements before and after synapse identity correction (*Figure 2c*). Note that double-innervated spines were always assumed to have one excitatory and one inhibitory component and were not included in the correction procedure. Finally, we used the corrected synapse density/fractions for the L5st group due to large spine-preferring input fraction onto their dendrite shafts (*Figures 5e* and *7b*, gray crosses are measurements before correction).

### Synapse size estimation

A random subset of shaft and spine (n = 41 per AD type) synapses were used to determine the synaptic interface area (*Figure 2d*, S1, PPC, V2 and ACC). For this, an expert annotator placed two edges along the longest dimension of the synapse and its approximate orthogonal direction. These two edges were used as minor and major axes of an ellipse to estimate the contact area ($area = \pi * semi-major\ axis\ * semi-minor\ axis$).

### Distribution of main bifurcations and axonal paths along upper cortex

We obtained the main bifurcation depth relative to pia by transforming the datasets into a common coordinate system as described above (n = 82, *Figure 1c*, main bifurcation annotations). Next, we used this to create a histogram (bin size of 20 µm) and probability density estimate (bandwidth: 25 µm) along the cortical depth (*Figure 3c*, left panel) for main bifurcation densities. Note that this is only a subset of main bifurcations within each image volume (See *Figure 1b*).

We used these datasets to also obtain a density estimate for inhibitory axonal paths. For this, we removed the nodes used for marking synapse locations in the inhibitory axonal reconstructions and computed the average fraction (across S1, V2, ACC and PPC datasets) of the remaining axonal path within 20 µm tangential cortical slices (*Figure 3c*, right panel).

## Conditional innervation probability of inhibitory axons

We selected a random subset of putative inhibitory synapses from layer 2 (n = 21, 20, 21, 30 for S1, V2, PPC and ACC) and deep layer (n = 19, 20, 20, 32 for S1, V2, PPC and ACC) ADs. An expert annotator (JO or AK) then reconstructed the axons throughout the dataset. We also annotated all the other synapses and postsynaptic partners for each axon. In addition, reconstructions were checked by another expert annotator (JO or AK) for morphological irregularities such as sharp branching angles and untraced endings.

The postsynaptic targets were categorized into one of the following: layer 2 or deep layer apical dendrite (shaft and spine of trunk and primary branches), shaft of other dendrites, single- or double-innervated (including neck targeting) spine, layer 2 cell body, axon initial segment (AIS) or glia. The innervation fraction of an axon was computed by dividing the number of synapses for each specific target by the total number of synapses of that axon (the seed synapse was excluded). The innervation fractions were averaged for each AD seed type (*Figure 3b,d*). Each dataset average was also computed separately (*Figure 4b–c*).

## Multiple innervation of an AD by inhibitory axons

To measure the frequency of multiple innervation of an AD by inhibitory axons used in our investigation of axonal specificity, we annotated all the cases were such multi-innervation happened (*Figure 3e*). We then measured the average number of synapses on each distinct AD target (*Figure 3f*). Axons with no synapses on a L2 (n = 18, 46 for L2 and DL seeded axons, respectively) or DL AD (n = 48, 24 for L2 and DL seeded axons, respectively) were excluded since averages are not defined on empty sets. To understand the effect of multiple innervation on AD type preference, we calculated the average fraction of AD targeting disregarding multiple innervation of the same target for axons seeded from either AD type (*Figure 3g*).

## Visualization of neurites and their synapses

To visualize the surface of the main bifurcation of PPC and S1 apical dendrites and axons, we segmented the image volume using SegEM (*Berning et al., 2015*) and collected all the segments comprising the dendritic shafts. The volume data was then imported to MATLAB, binarized and smoothed using a (9 × 9 × 9) voxel Gaussian convolution kernel with standard deviation of 8 voxels. The isosurface was then constructed at a threshold of 0.2 (*Figure 1c*).

Volume of the AD shaft at main bifurcation in V2 and ACC datasets were generated by tracing the outline of the dendrites using the volume tracing mode in webKnossos and 3D data was processed in a similar fashion (*Figure 1c*).

Skeleton reconstructions of axonal (*Figure 2a*) and dendritic paths (*Figures 1b,c*, *5a* and *7a*, *Figure 5—figure supplement 1*) were represented by tubes. In addition, spheres were added to represent input (*Figures 1c* and *7a*, *Figure 7—figure supplement 1a*) and output (*Figure 2a*) synapses.

The EM ultra-structure of an AD, synapses and cell bodies was demonstrated by electron microscopy images (PPC and ACC datasets) of their cross-sections (*Figures 1c* and *2a,e,f*).

The visualizations were generated using MATLAB (mathworks) and Amira (Thermo Fisher Scientific, USA).

### Statistics

Significance of difference between L2 and DL AD excitatory and putative inhibitory synapse densities (*Figures 1d,e* and *2g*), putative inhibitory fractions (*Figure 1f,g*), synapse size (*Figure 2d*), double innervation of spines (*Figure 2e*), AD diameter (*Figure 2f*), AD innervation fraction (*Figure 3b*) and L5 subtype comparisons (*Figure 5d*) were tested using the non-parametric Wilcoxon rank-sum test. In addition, the rank-sum test was used to test for the difference of putative inhibitory fraction and excitatory/putative inhibitory synapse densities between the main bifurcation and the distal tuft area

for each pyramidal cell type (*Figure 7c*). Finally, the difference in the putative inhibitory fraction between L2 (with L2MN), 3, 5tt and 5st apical dendrites (n = 4 independent samples) at the main bifurcation and on the distal tufts were tested using the non-parametric Kruskal-Wallis test to account for the small sample sizes (n = 12, 10, 7, 11 main bifurcations and n = 9, 7, 9, 6 distal tuft branches for L2, 3, 5tt, 5st, respectively. *Figures 5e* and *7b*) followed by Tukey's honestly significant difference post-hoc test.

To identify significant differences in the axonal innervation fractions, a bootstrapping test was used. The specificities of the two axonal groups were concatenated and 10,000 bootstrap resamples were drawn to match the number of axons seeded from L2 and DL ADs. The mean difference in each bootstrap resample between L2 and DL groups was compared to the sample mean difference. The p-value was defined as the fraction of bootstrap resamples which had a value more extreme compared to the sample mean. The significance threshold was set to 0.05 with Bonferroni correction for eight comparisons (*Figure 3d*).

We used a multivariate analysis of variance (MANOVA) test to measure the effect of cortical region on inhibitory axonal innervation probability for six postsynaptic targets (AD, soma, spine and shaft of other dendrites). Axon initial segment (AIS) and glia innervation fractions were excluded since they contained few synapses and resulted in singular matrices when MANOVA test was applied (0.8%, 0.04% of total synapses, respectively). The null hypothesis states that the differences between the mean fractional innervations of putative inhibitory axons in S1, V2, PPC and ACC is statistically insignificant. We followed the MANOVA test when the null hypothesis was rejected with multiple comparisons using Bonferroni corrected one-way ANOVA tests (*Figure 4b,c*). We then determined the target with differential fractional innervation between different cortical regions.

To investigate the relationship between synaptic composition at the main bifurcation and its distance to soma, single-term exponential models (with offset) of the form $y = c + a * e^{bx}$ were fit to the data from layers 2–5. The coefficient of determination ($R^2$) was computed as a goodness of fit measure. We also used $R^2$ to compare models explaining the relationship between thick-tuftedness (PC 1 of the four dimensional L5 neuron morphological feature space) and the putative inhibitory fraction at the main bifurcation. To understand the relationship between morphology and synapse density of L5 neurons, we fitted a linear model to individual synapse densities and a linear fraction model (form: $\frac{ax+b}{cx+d}$) to their putative inhibitory fraction.

## Acknowledgements

We thank Matthew Larkum, Johannes Letzkus and Idan Segev for discussions, Benedikt Staffler, Emmanuel Klinger, Manuel Berning, Alessandro Motta for computational and Jakob Straehle, Meike Schurr, Yunfeng Hua for experimental advice, and Alessandro Motta and Benedikt Staffler for contributing code. We thank M S E A Aly, L Bezzenberger, A B Brandt, B Heftrich, A C Rix, B L Stiehl, C Arras, C M Schumm, D E Celik, D J Goffitzer, J Buß, K M Trares, K Weber, L Buxmann, L Decker, L C R Kreppner, M S Kronawitter, N M Böffinger, N Plath, S M Bohne, S Reichel, T Engelmann, T Ernst, T Winkelmeier, V C Kalbert, K Kramer, L Präve, M Präve, N Berghaus, O J Brandt, S S Wehrheim for neurite reconstructions, Heiko Wissler, Susanne Babl, Lisa Bezzenberger, Alexander Brandt, Raphael Jakoby, Raphael Kneißl and Marc Kronawitter for annotator training and task management and Heiko Wissler for support with visualizations.

## Additional information

### Competing interests

Moritz Helmstaedter: Reviewing editor, *eLife*. The other authors declare that no competing interests exist.

## Funding

| Funder | Grant reference number | Author |
|---|---|---|
| Max-Planck-Gesellschaft | Open-access funding | Ali Karimi<br>Jan Odenthal<br>Florian Drawitsch<br>Kevin M Boergens<br>Moritz Helmstaedter |

The funders had no role in study design, data collection and interpretation, or the decision to submit the work for publication.

## Author contributions

Ali Karimi, Resources, Data curation, Software, Formal analysis, Validation, Investigation, Visualization, Methodology; Jan Odenthal, Data curation, Formal analysis, Validation, Visualization; Florian Drawitsch, Resources, Data curation, Validation, Investigation; Kevin M Boergens, Data curation, Investigation, Methodology; Moritz Helmstaedter, Conceptualization, Formal analysis, Supervision, Validation, Visualization

## Author ORCIDs

Ali Karimi (iD) https://orcid.org/0000-0002-6477-2523
Florian Drawitsch (iD) http://orcid.org/0000-0001-9543-1417
Moritz Helmstaedter (iD) https://orcid.org/0000-0001-7973-0767

## Ethics

Animal experimentation: All experimental procedures were performed according to the law of animal experimentation issued by the German Federal Government under the supervision of local ethics committees and according to the guidelines of the Max Planck Society. The experimental procedures were approved by Regierungspräsidium Darmstadt, under protocol ID V54 - 19c20/15 F126/1015 (LPtA, PPC-2) or V54 - 19 c 20/15 - F126/1002 (V2, PPC, ACC). The S1 sample was prepared following experimental procedures approved by Regierung von Oberbayern, 55.2-1-54-2532.3-103-12.

## Decision letter and Author response

Decision letter https://doi.org/10.7554/eLife.46876.sa1
Author response https://doi.org/10.7554/eLife.46876.sa2

# Additional files

## Supplementary files

• Supplementary file 1. Supplementary tables 1-5 reporting parameters for experiments and analysis, axonal preference for shaft and spine innervation, and location of pyramidal cell somata.

• Transparent reporting form

## Data availability

All 6 datasets are available for browsing at webknossos.org using the following links. S1: https://wklink.org/8732 V2: https://wklink.org/9812 PPC: https://wklink.org/1262 ACC: https://wklink.org/6712 LPtA: https://wklink.org/8912 PPC-2: https://wklink.org/6347 All software used for analysis is available at https://gitlab.mpcdf.mpg.de/connectomics/apicaltuftpaper (copy archived at https://github.com/elifesciences-publications/apicalTuftPaper) under the MIT license.

The following datasets were generated:

| Author(s) | Year | Dataset title | Dataset URL | Database and Identifier |
|---|---|---|---|---|
| Karimi A, Helmstaedter M | 2019 | PPC | https://wklink.org/1262 | WebKnossos, 1262 |

| Karimi A, Helm-staedter M | 2019 | PPC-2 | https://wklink.org/6347 | WebKnossos, 6372 |
| Odenthal J, Helm-staedter M | 2019 | ACC | https://wklink.org/6712 | WebKnossos, 6712 |
| Karimi A, Helm-staedter M | 2019 | V2 | https://wklink.org/9812 | WebKnossos, 9812 |
| Boergens KM, Helmstaedter M | 2019 | S1 | https://wklink.org/8732 | WebKnossos, 8732 |

The following previously published dataset was used:

| Author(s) | Year | Dataset title | Dataset URL | Database and Identifier |
|---|---|---|---|---|
| Drawitsch F, Helm-staedter M | 2018 | LPtA | https://wklink.org/8912 | WebKnossos, 8912 |

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
