## [Decision Letter]

**Acceptance summary:**

This study uses 3-D electron microscopy to map the inhibitory and excitatory synaptic input distribution of the apical dendrites of pyramidal cells in the mouse cortex. The authors compare excitatory/inhibitory synapse density in L2, L3 and L5 pyramidal cells in different cortical regions, at the bifurcation point from the apical dendritic trunk and the distal dendritic tufts. The resultant connectomic principles for the inputs onto pyramidal cells at their apical dendrites support specific computational properties of L2, L3 and subtypes of L5 pyramidal cells, and is a rich data set that should be of wide interest in the field.

**Decision letter after peer review:**

Thank you for submitting your article "Cell-type specific innervation of cortical pyramidal cells at their apical tufts" for consideration by *eLife*. Your article has been reviewed by three peer reviewers, and the evaluation has been overseen by a Reviewing Editor and Eve Marder as the Senior Editor. The reviewers have opted to remain anonymous.

The reviewers have discussed the reviews with one another and the Reviewing Editor has drafted this decision to help you prepare a revised submission.

Summary:

The reviewers were in the main positive about your study mapping the inhibitory and excitatory synaptic input distribution of the apical dendrites of pyramidal cells in the mouse cerebral cortex. Your comparison of excitatory/inhibitory synapse density in L2, L3 and L5 pyramidal cells in different cortical regions, at the bifurcation point from the apical dendritic trunk and the distal dendritic tufts, provides a very rich data set, one that should be of great interest to the field.

The reviewers found your results novel for the level of anatomical analysis and for implications of differential modes of inhibition for pyramidal cells in the cortex, including the following rules of synapse distribution along the apical dendrite: a. At the bifurcation point at the L1/2 border, inhibitory synapse density is higher in L2 cells than in deeper layer cells; b. At the distal apical tuft, the excitatory synapse density is lower in L5 cells than in L2 cells; c. Dendrites at the L1/2 border of L2 and deeper layer pyramidal cells are innervated by axons of different inhibitory cells. These results suggest that the computational properties of apical dendrites are distinct in L2 and L5 pyramidal cells.

Essential revisions:

Even with this praise, the reviewers had issues with the definition of the structures you analyzed, specifically, the criteria for classification of E and I synapses, for parsing orientation of distal dendrites, and for identifying axonal inputs. Many of these amendments can be textual, yet the first of these major points in particular, may affect your analysis.

1) Classification of synapses: Two of the reviewers at first assumed that you had classified synapses by the classical definition, i.e. symmetric as inhibitory and asymmetric as excitatory, but you state that "Shaft synapses, synapses on spine necks and secondary spine innervations were treated as inhibitory. Primary spine innervations were treated as excitatory synapses (Figure 1, 3)". a) Whereas one could conclude from the literature, that a single spine innervation represents an excitatory synapse, and a doubly innervated spine having one inhibitory and one excitatory input (although dual excitatory innervations have been observed previously), the reasoning goes astray for shaft synapses, which have been shown to represent both, with recent studies indicating that 15-20% of asymmetric synapses (presumably glutamatergic) are made on the dendritic shaft in the upper layers of cortex.

b) Figure 2E, F, G, indicates that some of axons innervating a shaft then also make a synapse on a spine without another input more frequently than on a spine with another input. Most spines with an inhibitory input usually have another excitatory input in the neocortex. The axons making a single synapse on spines may be thus be excitatory; please explain.

c) Validation of your classification system is requested: Discuss the differences between your assumptions and the literature, and the possible impact of your assumptions on the presented connectomic analysis. Do you or others have evidence that shaft synapses are gephyrin-positive, for example, beyond the Lascone study you cite (in BioRxiv)?

2) Identification of apical dendrites: a) You based this identification on radial direction and diameter. Whereas this may apply for deep dendrites, there might be some ambiguity for superficial cells. How exactly are L2 pyramids distinguished from other cell types in superficial layers? Can you provide quantitative characteristics of the dendrites analyzed, and possibly of some dendrites and their somata that were not considered to belong to pyramids? Such information would provide a baseline for future studies, and would avoid generalization and possible biases.

b) You assumed that dendritic branches from thicker dendrites around the L1/2 border represent bifurcation points from an apical dendritic trunk. L2 pyramidal cells were reconstructed down to their somata, but deeper layer pyramidal cells not. L5 thick-tufted pyramidal cells frequently have the bifurcation points in the center of L2/3. Is it possible that only L5 thin-tufted cells are analyzed in this study. This sampling bias may affect the conclusions.

c) The authors make no distinction between thick tufted L5 (possibly L5B) and slender layer 5 (possibly L5A). There are distinct functional differences between them. It would be interesting if the authors can distinguish them and quantify the proportion of inhibitory inputs onto these two types?

3) Distinction between L2 and L3 pyramidal neurons:

A classification of pyramidal cells based on the depth of the soma may not be reliable (e.g. L2 vs. L3). Both subtypes are possibly intermingled in the more superficial regions. Another concern might be that L2 cells without a main bifurcation are excluded according to the listed criteria, as well as branches of either type that run more obliquely or even parallel to the surface. Please comment on this, and provide more numeric data such as cell body position, size, possibly branch numbers, etc. for future reference, again, to avoid over-generalization of the difference between L2 and L3.

4) Contact by axons: a) For the axons that were reconstructed, were multiple contacts with the same dendrite or perhaps even the same neuron excluded? Did this occur?

b) When discussing the axon identity in the Results section, the authors report only the axons that targeted dendritic shafts or spines, and not axons that targeted neither. In Figure 1H and Supplementary Figure 1B, the question of whether all reported axons target contact at least one spine or shaft is important in order to better understand the results.

5) Synapse density: a) You report excitatory and inhibitory densities as a function of distance, and do not report the density per unit area, which is more informative.

b) The inhibitory synapse density around the border of L1/2 is different between L2 and L5 cells. The dendritic size around the border may be different between L2 and L5 cells. It would therefore be informative to show the relation between dendrite thickness and inhibitory synapse density of the dendrites at this border.

[Editors' note: further revisions were suggested prior to acceptance, as described below.]

Thank you for resubmitting your work entitled "Cell-type specific innervation of cortical pyramidal cells at their apical dendrites" for further consideration by *eLife*. Your revised article has been evaluated by Eve Marder as the Senior Editor, a Reviewing Editor and two peer reviewers.

Both reviewers appreciated the revisions on your study mapping the inhibitory and excitatory synaptic input distribution of the apical dendrites of pyramidal cells in the mouse cortex. Your comparison of excitatory/inhibitory synapse density in L2, L3 and L5 pyramidal cells in different cortical regions, at the bifurcation point from the apical dendritic trunk and the distal dendritic tufts, is a rich data set that should be of wide interest in the field. The manuscript has been improved but there are some remaining issues that need to be addressed before acceptance, as outlined below:

The reviewers still have a lingering concern: While they are receptive to the concept that if an axon maintains shaft synapses in more than 90% of its dendritic contacts, it likely represents an inhibitory axon, they and I suggest that you state this at the beginning of the Results section and that you did not use the classic symmetry in the pre- and postsynaptic densities to identify inhibitory synapses. The use of the term inhibitory synapses comes before you test this with seeding method, which makes the argument a bit circular. We suggest that you start by referring to shaft and spine synapses (under Results, subsection “Inhibitory to excitatory synapse ratio”), and refer to inhibitory synapses only once you have made their verification argument (subsection “Shaft vs. spine synapses”). In the latter section you can then present a plot that contains the corrected densities or fractions. Alternatively, you could first develop the argument of Figure 2A, B, and then present the data that are currently in Figure 1, maintaining the term inhibitory – though we would prefer 'putative inhibitory' over 'inhibitory'.

In the Discussion, you could also caution that identification of excitatory and inhibitory synapses using the classical asymmetrical or symmetrical criteria was difficult because the image quality of SBEM is slightly sub-par. New Figure 3 clearly indicates that some of axons innervating a shaft next make a synapse on a spine with (not without) another input more frequently than on a spine without another input. From the revised Figures 2-4, the readers can now easily understand the logic by which to differentiate the excitatory and inhibitory axons, not just based on the single postsynaptic structure, but on consideration of successive targets. Restating these points in the Discussion would strengthen your arguments.

On a positive note, reviewer 2 wondered whether the apical dendrites analyzed here belong to thick or slender tufted pyramidal cells in layer 5 because the two types of pyramidal cells significantly differ in function. Your new analysis has revealed their differences in spine density and this reviewer applauded this finding, and also found your quantitative analysis and conclusion that dendritic size is a key factor in determining inhibitory synapse density a strong contribution.

---

## [Author Response]

Essential revisions:[…] The reviewers had issues with the definition of the structures you analyzed, specifically, the criteria for classification of E and I synapses, for parsing orientation of distal dendrites, and for identifying axonal inputs. Many of these amendments can be textual, yet the first of these major points in particular, may affect your analysis.1) Classification of synapses: Two of the reviewers at first assumed that you had classified synapses by the classical definition, i.e. symmetric as inhibitory and asymmetric as excitatory, but you state that "Shaft synapses, synapses on spine necks and secondary spine innervations were treated as inhibitory. Primary spine innervations were treated as excitatory synapses (Figure 1, 3)". a) Whereas one could conclude from the literature, that a single spine innervation represents an excitatory synapse, and a doubly innervated spine having one inhibitory and one excitatory input (although dual excitatory innervations have been observed previously), the reasoning goes astray for shaft synapses, which have been shown to represent both, with recent studies indicating that 15-20% of asymmetric synapses (presumably glutamatergic) are made on the dendritic shaft in the upper layers of cortex.

We fully agree with the point raised by the reviewers that a single shaft or spine synapse can in principle be excitatory or inhibitory. 3D EM data, however, allows us to reconstruct the axons presynaptic at these synapses; which provides the targets of all other of the output synapses of these axons. Therefore, by reconstructing the presynaptic axons, we can quantify in how many cases a shaft synapse is made by an axon that in fact prefers spine innervations, and vice versa. We used the reconstructions of 430 presynaptic axons from synapses of the apical dendrites in layers 1 and 2 (new Figure 2A-C), and determined their other synaptic targets. The distribution of the fraction of single-innervated spine targets of these axons were almost binary (new Figure 2B) with axons preferring either shaft or spines of dendrites with the exception of axons innervating the shafts of L5 slender tufted ADs (new Figure 2B). Quantitatively, 91 of 92 axons seeded from the shafts of L2 ADs made at least 80% of their other output synapses onto shafts, as well (and 89 of 92 axons made at least 90% of their other synapses onto shafts). Similarly, 34 of 35 axons seeded from spines at L2 ADs made at least 80% of their other synapses as primary spine innervations. Numbers for synapses seeded at DL ADs were comparable.

We then calculated a corrected inhibitory fraction based on this connectivity criteria for classifying excitatory and inhibitory input and found it to be rather unchanged for most cell types (new Figure 2C). For L5 slender tufted ADs, synapse densities were corrected for the synapse identity (blue, new Figure 5E, Figure 7B grey crosses indicate paired uncorrected values) due to the substantial spine-preferring input onto their AD shaft (44-60%).

In summary, while for single synapses, the identification of spine vs. shaft targets with the synaptic classification of excitatory vs. inhibitory is error prone, the classification of axonal type by preference of their other local output synapses yields misclassification rates of less than 7% (for similar results in layer 4, see also (Motta et al., 2019, Motta et al., 2019); it appears that the axons innervating ADs as studied here show a stronger bimodal preference for either shafts or spines than those reconstructed in L4, (Motta et al., 2019)).

b) Figure 2E, F, G, indicates that some of axons innervating a shaft then also make a synapse on a spine without another input more frequently than on a spine with another input. Most spines with an inhibitory input usually have another excitatory input in the neocortex. The axons making a single synapse on spines may be thus be excitatory; please explain.

This was an unfortunate error in former Figure 2. The labels have been corrected in current Figure 3D, 4B-C (previously as Figure 2E, F, G, respectively). The shaft-preferring axons (inhibitory) innervate mainly spines with an additional (presumably excitatory) synapse (91.35% of 821 spine synapses). They only innervate a spine without a partner 8.65% of the time. See also the additional analyses in Figure 2A-C described above.

c) Validation of your classification system is requested: Discuss the differences between your assumptions and the literature, and the possible impact of your assumptions on the presented connectomic analysis. Do you or others have evidence that shaft synapses are gephyrin-positive, for example, beyond the Lascone study you cite (in BioRxiv)?

Our connectivity metric for the identification of axonal input type suggests a very strong shaft- vs. spine- preference for axons innervating most pyramidal neurons (new Figure 2, section a above). L5st neurons, however, did receive a significant amount of input from spine-preferring axons onto their shafts. We, therefore, corrected our synapse density measurement for this cell-type specific difference. We also reviewed a few studies which investigated the fraction of asymmetric and symmetric synapses (assumed to be excitatory and inhibitory, respectively) on shaft and spine of pyramidal neurons using 3D-EM:

1) (Chen et al., 2012) used a combination of ssTEM with Teal-Gephyrin 2P imaging. The authors found 26/26 asymmetric synapses on spines (no shaft asymmetric innervation) and 6/10 inhibitory synapses on shaft of the L2/3 pyramidal neuron dendrite (length: 30 µm). 3/10 inhibitory synapses were located on double-innervated spines and 1/10 single-innervated spine.

2) (White and Rock, 1980) studied the basal dendrites of the spiny stellate neurons in L4 of primary somatosensory cortex using serial section TEM (ssTEM). They found 100% of shaft and 3% (14/359) of spine synapses to be symmetric.

3) (White and Hersch, 1982) analyzed multiple segments of apical dendrites in L5/6 pyramidal neurons using ssTEM. They found 20% of shaft synapses to be asymmetric and only 0.6% of spine synapses to be symmetric (Table 1 in (White and Hersch, 1982)).

In conclusion, the degree to which the spine/shaft preference proxies for identification of excitatory and inhibitory synapses is dendrite type specific and shall be studied in the specific region of study either using an estimation method as described here or dense reconstruction of axons where most axon could be identified based on their single-innervated spine innervation fraction.

Notably, studies where dendrite types are not considered and a volume average of asymmetric and symmetric synapses is calculated could result in high degrees of symmetric input to shafts (up to 50%, See (Kwon, MerchánPérez et al., 2018, Santuy, Rodriguez et al., 2018)).

2) Identification of apical dendrites: a) You based this identification on radial direction and diameter. Whereas this may apply for deep dendrites, there might be some ambiguity for superficial cells. How exactly are L2 pyramids distinguished from other cell types in superficial layers? Can you provide quantitative characteristics of the dendrites analyzed, and possibly of some dendrites and their somata that were not considered to belong to pyramids? Such information would provide a baseline for future studies, and would avoid generalization and possible biases.

To disambiguate the morphology of the apical dendrites used in our analysis, we reconstructed the shaft morphology of all pyramidal neurons used in our cell-type analysis (new Figure 5, Figure 7, new Figure 5—figure supplement 1 contains a cell type gallery of all reconstructed ADs from LPtA and PPC-2 datasets). We also added diameter measurements (new Figure 2F). In addition, we specifically looked for pyramidal neurons with obliquely directed apical dendrites (marginal neurons) and measured their inhibitory input and compared it to other L2 pyramidal neurons (new Figure 5, Figure 5—figure supplement 1). In the two examples we found of L2 marginal pyramidal neurons, we did not find evidence that their input fraction was different from L2 pyramidal cells. For the distinction of L2 and L3 pyramidal cells, we again were able to use our larger EM datasets for clear differentiation (see cell type gallery in Figure 5—figure supplement 1).

b) You assumed that dendritic branches from thicker dendrites around the L1/2 border represent bifurcation points from an apical dendritic trunk. L2 pyramidal cells were reconstructed down to their somata, but deeper layer pyramidal cells not. L5 thick-tufted pyramidal cells frequently have the bifurcation points in the center of L2/3. Is it possible that only L5 thin-tufted cells are analyzed in this study. This sampling bias may affect the conclusions.

We reconstructed all the pyramidal neurons used in our cell-type analysis in the high- and low-resolution segments of LPtA and PPC-2 datasets down to their soma of origin for L2-5 neurons (new Figure 5—figure supplement 1). We found the AD morphology of the former L5 group much closer to L5B (thick-tufted, 7/10) neurons. Next, we identified six new L5 neurons with smaller soma and slender dendrite morphology (Larkman and Mason, 1990) (L5 slender tufted neurons or L5A, new Figure 5A, new Figure 5—figure supplement 1A). Next, we used hierarchical clustering to classify L5 neurons based on their soma diameter, AD trunk diameter, number of oblique dendrites and main bifurcation depth (new Figure 5B-C). In addition, these neurons had a substantially lower spine density (new Figure 5D). L5st neurons also seemed to have an increased inhibitory input fraction (15-31%) as compared to L5tt neurons in upper cortex (new Figure 7C).

c) The authors make no distinction between thick tufted L5 (possibly L5B) and slender layer 5 (possibly L5A). There are distinct functional differences between them. It would be interesting if the authors can distinguish them and quantify the proportion of inhibitory inputs onto these two types?

We thank the reviewers for the insightful suggestion – the additional analyses of

L5 slender tufted cells yielded an interesting specialization in input fraction (new Figure 5, Figure 7).

3) Distinction between L2 and L3 pyramidal neurons:A classification of pyramidal cells based on the depth of the soma may not be reliable (e.g. L2 vs. L3). Both subtypes are possibly intermingled in the more superficial regions. Another concern might be that L2 cells without a main bifurcation are excluded according to the listed criteria, as well as branches of either type that run more obliquely or even parallel to the surface. Please comment on this, and provide more numeric data such as cell body position, size, possibly branch numbers, etc. for future reference, again, to avoid over-generalization of the difference between L2 and L3.

We explicitly looked for L2 pyramidal neurons with an apical dendrite running oblique to the pial surface and found two of these marginal neurons (L2MN, new Figure 5, Figure 5—figure supplement 1A). We also compared their synaptic input composition with other L2 neurons (Results section, new Figure 5E, Figure 6A, Figure 7B). Our preliminary analysis (n=2) fails to rule out the possibility that L2MN neurons have a similar inhibitory fraction to other L2 neurons. For the distinction between L2 and L3, please see the new cell type gallery. Also, we report the relation between synaptic input composition and cell body depth, and AD length (Figure 6), which should avoid any overly binary distinctions.

4) Contact by axons: a) For the axons that were reconstructed, were multiple contacts with the same dendrite or perhaps even the same neuron excluded? Did this occur?

We analyzed the frequency of multiple contacts with ADs (new Figure 3E-G) and its effect on our conclusions. Multiple contacts within the EM volumes we analyzed occurred in 23% of the cases, such that the preference of axons for innervating certain types of ADs remained rather substantial even when disregarding the multiple innervations (Figure 3G).

b) When discussing the axon identity in the Results section, the authors report only the axons that targeted dendritic shafts or spines, and not axons that targeted neither. In Figure 1H and Supplementary Figure 1B, the question of whether all reported axons target contact at least one spine or shaft is important in order to better understand the results.

All axons reported in the manuscript were seeded from either spine or shaft synapses of an AD. Thus, they innervated at least the shaft or spine of the seed dendrite. For their remaining targets, (example: new Figure 2A), we identified these for a subset of axons, and report the distribution of axonal targets in Figure 3D and Figure 4 – but always for axons that targeted at least one spine or one shaft of an AD. We added a statement to that end to the Results subsection “Comparative analysis of innervation preference between cortex types”.

5) Synapse density: a) You report excitatory and inhibitory densities as a function of distance, and do not report the density per unit area, which is more informative.

We thank the reviewers for this suggestion. We additionally measured the diameter of apical dendrites and used it to calculate the synapse density per unit surface area of each cell type (new Figure 2F-G). All conclusions remained unaltered when normalizing for surface area of apical dendrites instead of their shaft path length.

b) The inhibitory synapse density around the border of L1/2 is different between L2 and L5 cells. The dendritic size around the border may be different between L2 and L5 cells. It would therefore be informative to show the relation between dendrite thickness and inhibitory synapse density of the dendrites at this border.

We added the relationship between the apical dendrite diameter and inhibitory/excitatory synapse density for apical dendrites of different cell types (new Figure 2—figure supplement 1). We found a correlation between inhibitory synapse density and the diameter of the apical dendrite. However, the diameter differences were not substantial enough to change the patterns of synapse density observed using the distance-based metric (new Figure 2G).

[Editors' note: further revisions were suggested prior to acceptance, as described below.][…] The manuscript has been improved but there are some remaining issues that need to be addressed before acceptance, as outlined below:The reviewers still have a lingering concern: While they are receptive to the concept that if an axon maintains shaft synapses in more than 90% of its dendritic contacts, it likely represents an inhibitory axon, they and I suggest that you state this at the beginning of the Results section and that you did not use the classic symmetry in the pre- and postsynaptic densities to identify inhibitory synapses. The use of the term inhibitory synapses comes before you test this with seeding method, which makes the argument a bit circular. We suggest that you start by referring to shaft and spine synapses (under Results, subsection “Inhibitory to excitatory synapse ratio”), and refer to inhibitory synapses only once you have made their verification argument (subsection “Shaft vs. spine synapses”). In the latter section you can then present a plot that contains the corrected densities or fractions. Alternatively, you could first develop the argument of Figure 2A, B, and then present the data that are currently in Figure 1, maintaining the term inhibitory – though we would prefer 'putative inhibitory' over 'inhibitory'.In the Discussion, you could also caution that identification of excitatory and inhibitory synapses using the classical asymmetrical or symmetrical criteria was difficult because the image quality of SBEM is slightly sub-par. New Figure 3 clearly indicates that some of axons innervating a shaft next make a synapse on a spine with (not without) another input more frequently than on a spine without another input. From the revised Figures 2-4, the readers can now easily understand the logic by which to differentiate the excitatory and inhibitory axons, not just based on the single postsynaptic structure, but on consideration of successive targets. Restating these points in the Discussion would strengthen your arguments.On a positive note, reviewer 2 wondered whether the apical dendrites analyzed here belong to thick or slender tufted pyramidal cells in layer 5 because the two types of pyramidal cells significantly differ in function. Your new analysis has revealed their differences in spine density and this reviewer applauded this finding, and also found your quantitative analysis and conclusion that dendritic size is a key factor in determining inhibitory synapse density a strong contribution.

We appreciate your comments and have made all changes to the description of synapses as requested:

- We start by describing shaft and spine synapses, only (subsection “Putative inhibitory to excitatory synapse ratio”);

- Switch to the term *putative inhibitory* after the calibration (subsection “Shaft vs. spine synapses” ff);

- Added a cautionary statement to the beginning of the Results;

- Added a paragraph to the Discussion (subsection “Distinction of excitatory vs. inhibitory synapses”).